# Neuronal network dysfunction in a model for Kleefstra syndrome mediated by enhanced NMDAR signaling

Monica Frega[1,2,7], Katrin Linda[1,7], Jason M. Keller[1], Güvem Gümüş-Akay[1], Britt Mossink[1], Jon-Ruben van Rhijn[3], Moritz Negwer[1], Teun Klein Gunnewiek[4], Katharina Foreman[3], Nine Kompier[3], Chantal Schoenmaker[1], Willem van den Akker[1], Ilse van der Werf [1,5], Astrid Oudakker[1], Huiqing Zhou[1,6], Tjitske Kleefstra[1], Dirk Schubert [3], Hans van Bokhoven[1,3] & Nael Nadif Kasri [1,3]*

Kleefstra syndrome (KS) is a neurodevelopmental disorder caused by mutations in the histone methyltransferase *EHMT1*. To study the impact of decreased EHMT1 function in human cells, we generated excitatory cortical neurons from induced pluripotent stem (iPS) cells derived from KS patients. Neuronal networks of patient-derived cells exhibit network bursting with a reduced rate, longer duration, and increased temporal irregularity compared to control networks. We show that these changes are mediated by upregulation of NMDA receptor (NMDAR) subunit 1 correlating with reduced deposition of the repressive H3K9me2 mark, the catalytic product of EHMT1, at the *GRIN1* promoter. In mice EHMT1 deficiency leads to similar neuronal network impairments with increased NMDAR function. Finally, we rescue the KS patient-derived neuronal network phenotypes by pharmacological inhibition of NMDARs. Summarized, we demonstrate a direct link between EHMT1 deficiency and NMDAR hyperfunction in human neurons, providing a potential basis for more targeted therapeutic approaches for KS.

[1] Department of Human Genetics, Radboudumc, Donders Institute for Brain, Cognition, and Behaviour, 6500 HB Nijmegen, Netherlands. [2] Department of Clinical neurophysiology, University of Twente, 7522 NB Enschede, Netherlands. [3] Department of Cognitive Neuroscience, Radboudumc, Donders Institute for Brain, Cognition and Behaviour, 6500 HB Nijmegen, Netherlands. [4] Department of Anatomy, Radboudumc, Donders Institute for Brain, Cognition and Behaviour, 6500 HB Nijmegen, Netherlands. [5] Laboratory of Nanotechnology for Precision Medicine, Italian Institute of Technology, Genova, Italy. [6] Department of Molecular Developmental Biology, Faculty of Science, Radboud Institute for Molecular Life Sciences, Radboud University, 6500 HB Nijmegen, Netherlands. [7] These authors contributed equally: Monica Frega, Katrin Linda. *email: n.nadif@donders.ru.nl

Advances in human genetics over the past decade have resulted in the identification of hundreds of genes associated with intellectual disability (ID) and autism spectrum disorder (ASD)[1]. Within this group of genes the number of chromatin regulators is remarkably high[2–4]. These ASD/ID-linked chromatin regulators are engaged in genome-wide covalent DNA modifications, posttranslational modifications of histones, and control of genomic architecture and accessibility[5] to control the expression of genes important for neurodevelopment and/or neuroplasticity[3]. Nevertheless, there is still a large gap between elucidating the genetic architecture of neurodevelopmental disorders (NDDs) and deciphering the cellular or molecular pathobiology[6]. In particular, we require better understanding of the relevance of genetic changes with respect to downstream functional consequences and whether there is overlap between patients within the clinical spectrum[6].

Kleefstra syndrome (KS) (OMIM#610253) is an example of a rare monogenic NDD with ID, ASD, hypotonia, and dysmorphic features[7–9]. KS is caused by heterozygous de novo loss-of-function mutations in the *EHMT1* gene (euchromatin histone lysine methyltransferase 1) or by small 9q34 deletions harboring the gene[7]. In a complex with EHMT2, EHMT1 methylates histone 3 at lysine 9 (H3K9me1 and H3K9me2), promoting heterochromatin formation leading to gene repression[10]. Constitutive and conditional loss of EHMT1 function in mice and *Drosophila* lead to learning and memory impairments[11–13]. In addition, $Ehmt1^{+/−}$ mice recapitulate the developmental delay and autistic-like behaviors that are observed in KS patients[14,15]. At the cellular level, these mice show a significant reduction in dendritic arborization and number of mature spines in CA1 pyramidal neurons[11]. The dynamic regulation of H3K9me2 by EHMT1/2 is also involved in synaptic plasticity and learning and memory[16–18]. EHMT1 and 2 are required for synaptic scaling, a specific form of homeostatic plasticity, by regulating the expression of brain-derived neurotrophic factor (BDNF)[16]. Yet it remains unknown how deficits caused by loss of EHMT1 mechanistically affect the development of human neuronal networks.

Human-induced pluripotent stem (iPS) cell technology[19] enables us to the study the specific role of individual cell types in developing neural circuits. Patient-derived neurons allow us to examine the early pathophysiology of NDDs using single-cell and neuronal network electrophysiological recordings to recapitulate disease progression[20,21].

Here, we generated iPS cell lines from three patients with different *EHMT1* loss-of-function mutations to differentiate them into excitatory cortical neurons. Through in-depth characterization at single-cell and neuronal network level, we uncovered a robust and defined phenotype that was consistent across all patient lines and was also observed in neurons with CRISPR-engineered disruption of *EHMT1*. At the molecular and cellular level, we show that the electrophysiological phenotype is mediated by upregulation of NMDA receptor (NMDAR) subunit 1, which we also find in $Ehmt1^{+/−}$ mice. We conclude by showing that pharmacological inhibition of NMDARs rescues the KS-associated network phenotypes. Therefore, our findings establish a direct link between EHMT1 deficiency in human neurons and NMDAR hyperfunction, providing new insights into the pathophysiology of KS.

## Results

**Single-cell level characterization of KS neurons**. We generated iPS cell lines from two patients with KS and two healthy subjects (respectively $KS_1$, $KS_2$ and $C_1$, $C_2$) (Fig. 1a, Supplementary Fig. 1, Material and Methods). One patient ($KS_1$) had a frameshift mutation in *EHMT1* leading to a premature stop codon (p. Tyr1061fs, patient 25 in ref. [22]), while the other patient had a missense mutation in the Pre-SET domain (p.Cys1042Tyr, patient 20 in ref. [8]), predicted to disrupt the conformation of this domain. As expected Western blot and real-time quantitative polymerase chain reaction (RT-qPCR) analyses revealed a 50% reduction of EHMT1 expression in $KS_1$, while $KS_2$ showed normal EHMT1 expression levels (Fig. 1b, Supplementary Fig. 2A). In addition to these lines, iPS cells were generated from an individual who has a mosaic microdeletion on chromosome 9q34 (233 kb) including *EHMT1*[23]. We selected an iPS clone harboring the KS-causing mutation ($KS_{MOS}$) as well as a control clone not carrying the *EHMT1* deletion ($C_{MOS}$) (Fig. 1a, Supplementary Figs. 1 and 2). This isogenic pair shares the same genetic background except for the KS-causing mutation, thereby reducing variability and enabling us to directly link phenotypes to heterozygous loss of *EHMT1*. Western blot analysis and RT-qPCR analysis showed a 40% reduction of EHMT1 expression in $KS_{MOS}$ compared to $C_{MOS}$ (Fig. 1b, Supplementary Fig. 2A). All selected clones showed positive expression of pluripotency markers (OCT4, TRA-1-81, NANOG, and SSEA4) and single-nucleotide polymorphism (SNP) arrays were performed to confirm genetic integrity (Supplementary Fig. 1, Material and Methods).

We differentiated iPS cells into a homogeneous population of excitatory upper layer cortical neurons (iNeurons) by forced expression of the transcription factor transgene *Ngn2*[24]. For all experiments, iNeurons were co-cultured with freshly isolated rodent astrocytes[25] to facilitate neuronal and network maturation (Fig. 1c). All iPS lines were able to differentiate into MAP2-positive neurons at similar efficiency (Supplementary Fig. 2C, D). EHMT1 expression was reduced in neurons derived from $KS_1$ and $KS_{MOS}$, but not $KS_2$ (Supplementary Fig. 2B) compared to controls. However, all KS iNeurons showed reduced H3K9me2 immunoreactivity, indicative of EHMT1 haploinsufficiency (Supplementary Fig. 2E). Twenty-one days after the start of differentiation (days in vitro, DIV), both, control and KS iNeurons showed mature neuronal morphology. We measured this by reconstruction and quantitative morphometry of DsRed-labeled iNeurons. We observed no significant differences between control and KS iNeurons in any aspect of neuronal somatodendritic morphology, including the number of primary dendrites, dendritic length and overall complexity (Fig. 1d, Supplementary Fig. 3A–F).

ID and ASD have been associated with synaptic deficits in rodents and humans[21,26]. We therefore investigated whether *EHMT1*-deficiency leads to impairments in synapse formation. To this end, we stained control and KS iNeurons for pre- and postsynaptic markers (i.e., synapsin1/2 and PSD95, respectively) at DIV 21. We observed that putative functional synapses were formed on control and KS iNeurons (Supplementary Fig. 3G), without any indications for differences in the number of synaptic puncta between the different iPS cell lines (Fig. 1e, Supplementary Fig. 3G). Furthermore, whole-cell patch-clamp recordings at DIV 21 of iNeurons grown in the presence of tetrodotoxin (TTX) also revealed no differences in the frequency or amplitude of AMPA receptor (AMPAR)-mediated miniature excitatory postsynaptic currents (mEPSCs, Supplementary Fig. 3H). This indicates that KS iNeurons are generally not impaired in the AMPAR component of the excitatory input they receive (Fig. 1f, Supplementary Fig. 3H). At DIV 21 nearly 90% of control and KS patient iNeurons fired multiple APs, indicative of a mature state of their electrophysiological properties (Supplementary Fig. 3I). When recording intrinsic passive and active properties from control and KS patient iNeurons at DIV 21, we found no differences when comparing controls with KS lines. However, we did observe minor differences in AP decay time and Rheobase for

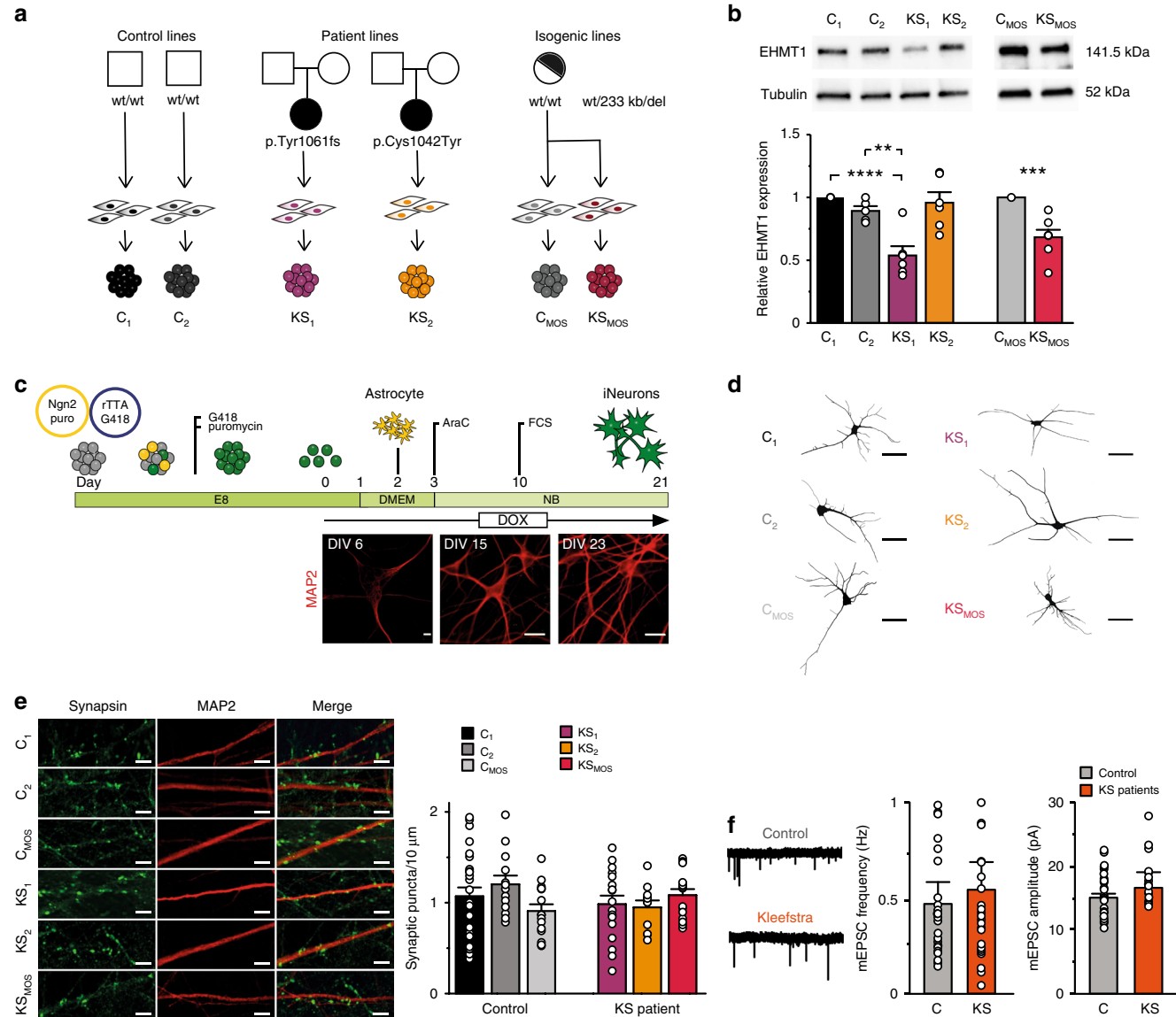

**Fig. 1** Generation and characterization of iPS cell-derived neurons from KS patients. **a** KS and control iPS lines used in this study. **b** Western blot and quantification of EHMT1 protein levels in iPS cells, $n = 6$–7 for each condition. **c** Schematic presentation of the differentiation protocol, including representative images of control iNeurons immunostained for MAP2 during development (scale bar 10 μm). **d** Representative somatodendritic reconstructions of control and KS iNeurons (scale bar 50 μm). **e** Representative images of control and KS iNeurons stained for MAP2 (red) and synapsin 1/ 2 (green) at DIV 21 (scale bar 5 μm) and quantification of synapsin puncta, $n = 25$ for $C_1$; $n = 13$ for $C_2$; $n = 15$ for $C_{MOS}$; $n = 17$ for $KS_1$; $n = 11$ for $KS_2$; $n = 15$ for $KS_{MOS}$. **f** Representative example traces of mEPSCs from control and KS iNeurons at DIV 21. Quantification of the frequency and amplitude of mEPSCs in control ($C_1$, $C_{MOS}$, and $C_{CRISPR}$) and KS ($KS_1$, $KS_{MOS}$, and $KS_{CRISPR}$) iNeurons (i.e., pooled value for control and KS iNeurons), $n = 30$ for C, $n = 29$ for KS. Data represent means ± SEM. **P < 0.005, ***P < 0.0005, ****P < 0.0001, one-way ANOVA test and post hoc Bonferroni correction was performed between controls and KS iNeurons and Mann–Whitney test was performed between two groups. Source data is available as a Source Data file

the specific lines KS2 and KS1, respectively. These differences were not observed between the isogenic lines (Supplementary Fig. 3J–N, Supplementary Data 1). Collectively, our data indicate that there were no significant differences between control and KS patient iNeurons with regard to neuronal morphology, excitatory synapses and intrinsic properties.

**KS neuronal networks show an aberrant pattern of activity.** Dysfunction in neuronal network dynamics has been observed in the brain of patients with psychiatric and neurological conditions[27]. In addition, neuronal network dysfunction has been identified in model systems for several ID/ASD syndromes[20,28]. Therefore, despite the normal properties of KS iNeurons on single

cell level, we hypothesized that impairments during brain development caused by loss of EHMT1 would be reflected at the network level. To test this hypothesis, we examined and compared the spontaneous electrophysiological population activity of neuronal networks derived from controls and KS patients growing on microelectrode arrays (MEAs) (Fig. 2a). MEAs allow us to noninvasively and repeatedly monitor neuronal network activity through extracellular electrodes located at spatially separated points across the cultures.

First, we monitored the in vitro development of control neuronal networks on MEAs. We found that the activity pattern of control iNeuron networks changed progressively over several weeks (Fig. 2b–f), similarly to what was observed in rodent neuronal cultures[29]. In particular, during the first three weeks of

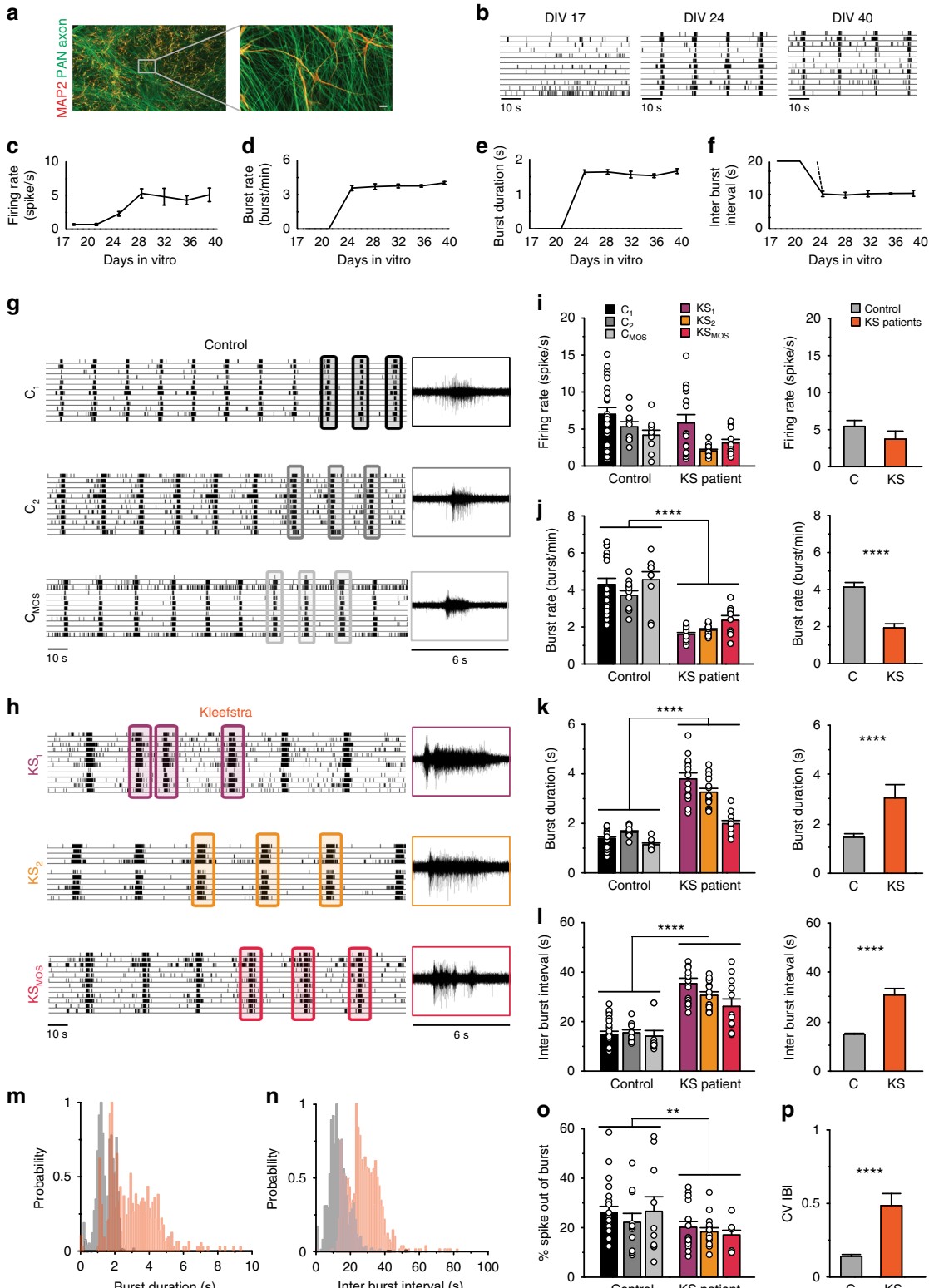

**Fig. 2** Spontaneous electrophysiological activity of control- and KS patient-derived neuronal networks. **a** Representative image of a control-derived neuronal network on MEAs stained for MAP2 (red) and PAN Axon (green). **b** Representative raster plot of electrophysiological activity exhibited by control-derived neuronal network at different time points during development. **c–f** Quantification of network properties as indicated. **g, h** Raster plot of spontaneous electrophysiological activity exhibited by **g** control and **h** KS networks at DIV 28. 6 s of raw data showing a burst recorded from a representative channel are shown for each iPS line. **i–l** Quantification of network parameters as indicated. **m, n** Histogram showing the distribution of **m** the network burst duration (bin size = 100 ms) and **n** the network inter burst interval (bin size = 1 s). **o** Quantification of percentage of spike outside network burst and **p** quantification of coefficient of variability of the inter-burst interval. $n = 23$ for $C_1$; $n = 10$ for $C_2$; $n = 10$ for $C_{MOS}$; $n = 15$ for $KS_1$; $n = 15$ for $KS_2$; $n = 12$ for $KS_{MOS}$. Data represent means ± SEM. $^*P < 0.05$, $^{**}P < 0.005$, $^{****}P < 0.0001$, one-way ANOVA test and post hoc Bonferroni correction was performed between controls and KS patient-derived cultures and Mann–Whitney test was performed between two groups. Source data is available as a Source Data file

differentiation, control iNeurons mainly showed sporadic APs, i.e., spikes (Fig. 2b–f, DIV 17), indicating they were immature and not yet integrated in a network. By week four electrical activity was organized into rhythmic, synchronous events (network burst), composed of many spikes occurring close in time and across all electrodes (Fig. 2b–f, DIV 24). This indicates that the iNeurons had self-organized into a synaptically connected, spontaneously active network. Both the firing rate, which is a general measure of total activity across the entire network, and network burst activity increased during development but plateaued by DIV 28 (Fig. 2c, d). After this time point neuronal network activity remained stable (Fig. 2c–f). We observed no difference in the overall level or pattern of activity between the $C_1$, $C_2$, and $C_{MOS}$ at DIV 28 (Fig. 2g, i–l). The highly reproducible network characteristics observed across all controls provided us with a consistent, robust standard to which we could directly compare KS-patient derived neuronal networks.

Next, we characterized the neuronal networks derived from KS patients. Similar to controls, network burst activity appeared by the fourth week in vitro (Fig. 2h). The global level of activity of KS networks was similar to controls (i.e., the firing rate, Fig. 2i). However, we found that network bursts occurred at lower frequency and with longer duration (Fig. 2j, k, m, Supplementary Fig. 4A, B). As a consequence of the lower network burst frequency, the inter-burst interval was longer (Fig. 2l, n, Supplementary Fig. 4C, D). We also observed that spike organization differed from controls, which was indicated by the smaller percentage of spikes occurring outside the network bursts (Fig. 2o). A final aspect is that KS networks also exhibited an irregular network-bursting pattern, illustrated by the statistically larger coefficient of variation (CV) of the inter-burst interval (Fig. 2p). Interestingly, the increased burst duration phenotype observed at the network level was also present at single-cell level (Supplementary Fig. S3O). Indeed, whole-cell voltage–clamp recordings of spontaneous excitatory postsynaptic currents (sEPSCs), showed that the activity received by KS-derived neurons was composed by longer burst durations than in control. Taken together, our data show that KS neuronal networks consist of fewer and irregular network bursts, and the bursts themselves were longer in duration than in control networks.

**CRISPR/Cas9 deletion of EHMT1 recapitulates KS phenotypes.** To further address whether heterozygous loss of *EHMT1* is causing the observed KS patient-derived network phenotypes, we expanded our analysis and generated a second set of isogenic human iPS cells. We made use of CRISPR/Cas9 gene editing technology to generate an isogenic control and *EHMT1* mutant iPS cell line with a premature stop codon in exon 2 ($C_{CRISPR}$ and $KS_{CRISPR}$, Fig. 3a, Supplementary Fig. 5F, G). Western blot and RT-qPCR analysis revealed that EHMT1 expression was significantly reduced in $KS_{CRISPR}$ iPS and iNeurons compared to $C_{CRISPR}$ (Fig. 3b, e, Supplementary Fig. 5F, G). Both, $C_{CRISPR}$ and $KS_{CRISPR}$ iPS cells differentiated equally well to iNeurons (Supplementary Fig. 5H). Furthermore, $KS_{CRISPR}$ iNeurons showed reduced H3K9me2 immunoreactivity compared to $C_{CRISPR}$ iNeurons (Supplementary Fig. 5I). We observed no differences in the formation of functional synapses, based on immunocytochemistry and mEPSC recordings between $C_{CRISPR}$ and $KS_{CRISPR}$, corroborating our results with the other KS cell lines (Fig. 3c, f, Supplementary Fig. 3H). At the network level, $C_{CRISPR}$ showed a control-like network phenotype (Fig. 3d, g–k). $KS_{CRISPR}$ networks exhibited a phenotype similar to the other KS patient networks with less frequent network bursts, longer duration and in an irregular pattern. This establishes a causal role for *EHMT1* in the observed neuronal network phenotypes.

Our results demonstrated that EHMT1 deficiency causes a reproducible neuronal network phenotype. We observed only nonsignificant iPS cell line-specific variability in the functional network properties of, both, the control and KS groups, so that the multiple descriptive parameters extracted from the raw MEA recordings clearly delineated control from KS networks. This was confirmed in an unbiased discriminant analysis of network parameters, where control and KS networks clearly clustered away from each other. This separation was not observed when the analysis was performed on single-cell parameters (i.e., morphology and intrinsic properties, Supplementary Fig. 6A–F). Our direct comparison of iNeurons derived from iPS cells with a frameshift, missense or deletion in *EHMT1* showed that the phenotype is due to aberrant EHMT1 enzymatic activity rather than the disrupted protein.

**KS iNeurons show increased sensitivity to NMDAR antagonists.** KS patient-derived neuronal networks showed an aberrant pattern of activity, mainly characterized by network bursts with longer durations than controls. Previous studies on rodent-derived neuronal networks have shown that burst duration is directly influenced by AMPARs and NMDARs. Specifically, previous reports used receptor-type specific antagonists to show that AMPAR-driven bursts have short durations while NMDAR-driven bursts have comparatively longer durations[30]. We therefore hypothesized that increased NMDAR activity contributed to the lengthened bursts in KS networks. To test this, we pharmacologically blocked either AMPARs or NMDARs and compared the effect on control and KS neuronal network activity at DIV 28. In accordance with previous work[30,31], we found that acute treatment with an AMPAR antagonist (NBQX, 50 μM) abolished all network burst activity, whereas inhibiting NMDARs (D-AP5, 60 μM) only slightly decreased the network burst activity (Fig. 4a, c) for control networks. This indicated that network burst activity is mainly mediated by AMPARs. In particular, we found it to be mediated by GluA2-containing AMPARs, since the network bursts were not blocked with Naspm (10 μM), an antagonist that selectively blocks GluA2-lacking AMPARs (Supplementary Fig. 7B, pre-D-AP5). Similar to controls, in KS networks NBQX completely abolished network burst activity (Fig. 4b, d). However, in stark contrast to controls, D-AP5 robustly suppressed the bursting activity in EHMT1 deficient lines (Fig. 4b, d, Supplementary Fig. 7A). Interestingly, the suppression of network burst activity by D-AP5 in KS networks was transient. Network burst activity showed ~50% recovery after 30 min and had returned to baseline (i.e., pre-D-AP5 levels) after 24 h (Fig. 4e, Supplementary Fig. 7C). The early stages of homeostatic plasticity, specifically synaptic upscaling, are initiated by global neuronal inactivity and characterized by insertion of GluA2-lacking AMPARs (i.e., $Ca^{2+}$-permeable AMPARs, CP-AMPARs) into the synapse to restore activity[32]. To determine the nature of the recovery, we first blocked KS networks with D-AP5, and added Naspm after 1 h. Naspm completely blocked the reinstated network bursting activity, indicating the recovery in KS networks was due to synaptic insertion of GluA2-lacking AMPARs (Fig. 4e, Supplementary Fig. 7B). Of note, although in control networks burst activity was not suppressed by D-AP5, we did observe that Naspm had a small but significant effect on the burst rate. This indicates that NMDAR blockage in controls also induced less pronounced synaptic insertion of GluA2-lacking AMPARs. After 24 h, the network activity in KS networks was again completely suppressed with NBQX but only partially with Naspm, suggesting that GluA2-lacking AMPARs were actively exchanged with GluA2-containing AMPARs (Supplementary Fig. 7C). Collectively, our results demonstrate that NMDAR inhibition with

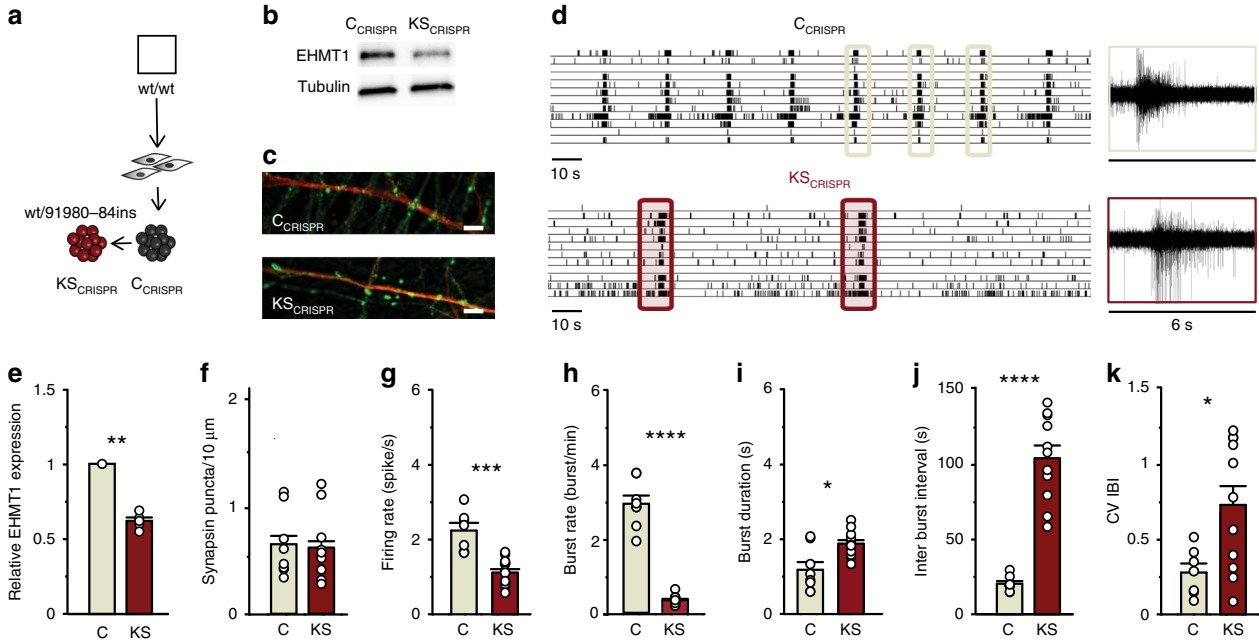

**Fig. 3** Spontaneous electrophysiological activity of neuronal network derived from control- and CRISPR/Cas9-edited iPS cells. **a** Isogenic line: $C_{CRISPR}$ and $KS_{CRISPR}$. **b** Western blot showing the EHMT1 protein levels in iPS cells. **c** Representative images of iNeurons stained for MAP2 (red) and synapsin 1/2 (green) at DIV 21 (scale bar 5 µm). **d** Representative raster plots showing spontaneous activity exhibited by $C_{CRISPR}$ and $KS_{CRISPR}$ at DIV 28 on MEAs. Totally, 6 s of raw data showing a burst recorded from a representative channel are shown. **e** Quantification of relative EHMT1 protein level, $n = 5$. **f** Quantification of synapsin puncta in $C_{CRISPR}$ and $KS_{CRISPR}$ iNeurons, $n = 9$ for $C_{CRISPR}$; $n = 13$ for $KS_{CRISPR}$. **g–k** Quantification of network parameters as indicated, $n = 7$ for $C_{CRISPR}$; $n = 12$ for $C_{CRISPR}$. Data represent means ± SEM. $*P < 0.05$, $**P < 0.005$, $***P < 0.0005$, $****P < 0.0001$, one-way ANOVA test and post hoc Bonferroni correction was performed between controls and KS networks and Mann–Whitney test was performed between two groups. Source data is available as a Source Data file

D-AP5 induces synaptic plasticity in KS networks, allowing reinstating network burst activity through the incorporation of GluA2-lacking AMPARs, which later on are replaced by GluA2-containing AMPARs.

Intrigued by these findings, we sought an alternative way to block network bursting. The classic method for inducing synaptic upscaling, with the sodium channel blocker TTX[33], would necessarily prevent observation of early stages of the dynamic recovery on MEAs. To circumvent this issue, we used the antiepileptic drug Retigabine, which is a voltage-gated $K^+$-channel ($K_v7$) activator that effectively hyperpolarizes the resting membrane potential in neurons[34]. We reasoned that Retigabine would have the combined effect of acutely hampering neurons in reaching AP threshold while leaving the voltage-gated $Na^+$ channels unaffected. Thus, while simultaneously strengthening the $Mg^{2+}$ block on NMDARs, we could still observe any (re) occurring network activity on the MEA. Indeed, when we applied Retigabine (10 µM) to the networks, they were temporarily silenced, with no discernible spiking or bursting activity. Within 100 min, there was again a Naspm-sensitive recovery mediated by GluA2-lacking AMPARs occurring in control and KS networks, with an identical pattern (Supplementary Figs. 7D and 8A). This reinforced the notion that the electrophysiological differences we observed earlier between control and KS networks are a direct consequence of NMDAR activity in the KS iNeurons. Furthermore, we found that Retigabine treatment in KS networks decreased the burst length (Supplementary Figs. 7D and 8A), suggesting that Retigabine-induced plasticity allowed KS networks to switch from an NMDAR-dependent to AMPAR-driven bursting, similar to controls.

**NMDARs are upregulated in KS iNeurons.** The results from our pharmacological experiments suggested that NMDAR expression might be increased in KS iNeurons relative to controls. To test this hypothesis, we measured the transcripts of the most common NMDAR and AMPAR subunits by RT-qPCR for $C_{MOS}$ and $KS_{MOS}$ iNeurons (Supplementary Fig. 7E). We were intrigued to find a fourfold upregulation of *GRIN1* mRNA, which encodes NMDAR subunit 1 (NR1), the mandatory subunit present in functional NMDARs. We found no significant changes in any other NMDAR (*GRIN2A*, *GRIN2B*, *GRIN3A*) or AMPAR (*GRIA1*, *GRIA2*, *GRIA3*, and *GRIA4*) subunit that we analyzed. We further corroborated these results with Western blot analysis, which revealed significantly increased NR1 expression for $KS_{MOS}$ and $KS_{CRISPR}$ iNeurons compared to $C_{MOS}$ and $C_{CRISPR}$ (Fig. 5a, Supplementary Data 1). Our previous functional data indicated that the reduction in methyltransferase activity of EHMT1 was directly responsible for the phenotypes we observed at the network level. Therefore, we used chromatin immunoprecipitation qPCR (ChIP-qPCR) to investigate whether the increased *GRIN1* expression correlated with reduced H3K9me2 at the *GRIN1* promoter. Our results showed that for $KS_{MOS}$ and $KS_{CRISPR}$ iNeurons H3K9me2 occupancy was reduced at the *GRIN1* promotor (Fig. 5b). In accordance with our previous study in *Ehmt1*$^{+/-}$ mice[16], we also found that the occupancy at the *BDNF* promoter was reduced in $KS_{MOS}$ and $KS_{CRISPR}$ iNeurons (Fig. 5b). Next, using immunocytochemistry, we found that NR1 was significantly increased in $KS_{MOS}$ and $KS_{CRISPR}$ compared to $C_{MOS}$ and $C_{CRISPR}$ iNeurons (Fig. 5c). Finally, we investigated whether the increased NR1 level also resulted in increased synaptic NMDAR activity. To this end we infected control or KS iNeurons at 7 DIV with an adeno-associated virus (AAV2) expressing mCherry-tagged channelrhodopsin (±80% infection efficiency). We recorded from uninfected cells in voltage clamp at a holding potential of −70 mV (AMPAR) or +40 mV (NMDAR) and measured (blue)light-evoked synaptic responses by exciting

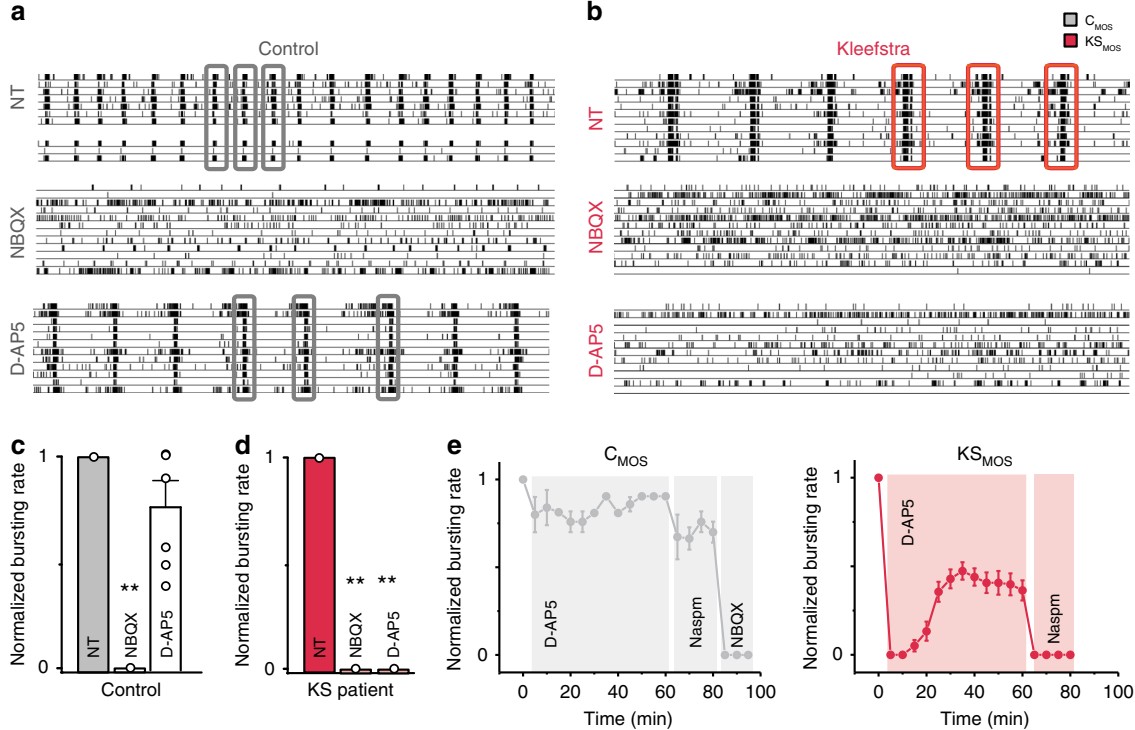

**Fig. 4** Effect of AMPA- versus NMDA-receptor blockade on control and KS patient-derived neuronal network activity. **a**, **b** Representative raster plots showing 3 min of spontaneous activity from **a** control ($C_{MOS}$) and **b** KS ($KS_{MOS}$) neuronal networks grown on MEAs at DIV 28. Where indicated, the cells were either non-treated (NT) or treated with 50 μM NBQX to block AMPA receptors or 60 μM D-AP5 to block NMDA receptors. **c**, **d** Bar graphs showing the effect of NBQX and D-AP5 on the network burst rate for **c** control and **d** KS neuronal networks ($C_{MOS}$ and $KS_{MOS}$ respectively). The values are normalized to the NT data ($n = 6$ for $C_{MOS}$ and $n = 6$ for $KS_{MOS}$ in all conditions). **e** The effect of a 1-hour D-AP5 treatment on $C_{MOS}$ and $KS_{MOS}$ neuronal network activity. After 1 h, the calcium-permeable AMPA receptor blocker Naspm (10 μM) or NBQX were added. Data represent means ± SEM. **$P < 0.005$, ***$P < 0.0005$, $n = 8$ for $C_{MOS}$ and $n = 9$ for $KS_{MOS}$, one-way ANOVA test followed by a post hoc Bonferroni correction was performed between conditions. Source data is available as a Source Data file

nearby channelrhodopsin-expressing cells (mCherry positive) (Fig. 5d). The NMDAR/AMPAR ratio showed to be significantly increased in $KS_{MOS}$ and $KS_{CRISPR}$ iNeurons (Fig. 5e). This increase in NMDAR/AMPAR ratio is likely due to an increased NMDAR activity since frequency and amplitude of AMPAR-mediated mEPSCs remained unchanged (Supplementary Fig. 3H).

Taken together, our data show that the loss of EHMT1 activity in KS lines results in a reduction of the repressive H3K9me2 mark, causing an upregulation of NR1 and increased synaptic NMDAR activity.

**Altered neuronal network activity in $Ehmt1^{+/-}$ mice.** Having established that EHMT1-deficiency alters neuronal network activity due to NR1 upregulation in KS iNeurons, we next set out to measure neuronal network activity in $Ehmt1^{+/-}$ mice, a validated mouse model that recapitulates the core features of KS[14,15]. Similar to what we found in iNeurons, primary cultures of cortical neuronal networks derived from $Ehmt1^{+/-}$ mice showed network bursts with lower frequency and longer duration compared to cultures from littermate controls. The MFR was unaltered (Fig. 6a). Using whole-cell voltage clamp recordings in acute brain slices, we measured the ratio between AMPAR- and NMDAR-mediated currents from cortical Layer 4 to Layers 2/3 synapses. We found that the NMDAR/AMPAR ratio was significantly increased in cortical networks of $Ehmt1^{+/-}$ mice compared to WT littermates (Fig. 6b). We found no changes in the kinetics of NMDAR-mediated currents, suggesting that there is no difference in the expression of NMDAR subunits 2A or 2B between WT and $Ehmt1^{+/-}$ mice (Supplementary Fig. 6B)[35].

Finally, we found no changes in the frequency or amplitude of AMPAR-mediated mEPSCs suggesting that the increased NMDAR/AMPAR ratio in $Ehmt1^{+/-}$ mice is due to increased NMDAR activity (Fig. 6c).

**NMDAR inhibition rescues KS neuronal network phenotypes.** Our previous experiments showed that by inhibiting NMDARs in KS networks for 24 h, we could shift the balance so that neuronal networks were progressively driven by GluA2-containing AMPARs, similar to controls (Supplementary Fig. 7C). Based on these results, we reasoned that the phenotypes in KS networks could be rescued by chronically inhibiting NMDARs in mature neuronal networks. We chose to potently block the channel pore of NMDARs, and thereby primarily inhibiting postsynaptic calcium flux, with the selective, noncompetitive open-channel blocker MK-801[36]. The immediate effects of MK-801 (1 μM) on KS network activity were similar to D-AP5, where network bursting was transiently suppressed and then recovered by insertion of GluA2-lacking AMPARs, which were again completely inhibited by Naspm (Fig. 7a, Supplementary Fig. 8B). Next, we treated $C_{MOS}$ and $KS_{MOS}$ networks beginning at DIV 28 for 7 days with MK-801. We found that chronically blocking NMDARs in $KS_{MOS}$ networks reversed the major network parameters that we measured in $KS_{MOS}$ iNeurons, bringing them closer to controls (Fig. 7b–g). More specifically, we observed an increased burst frequency (Fig. 7d), with a parallel reduction in the inter-burst interval (Fig. 7f). Furthermore, the network burst duration (NBD) was reduced (Fig. 7e) and the network bursting became more regular (Fig. 7g) showing values similar to control,

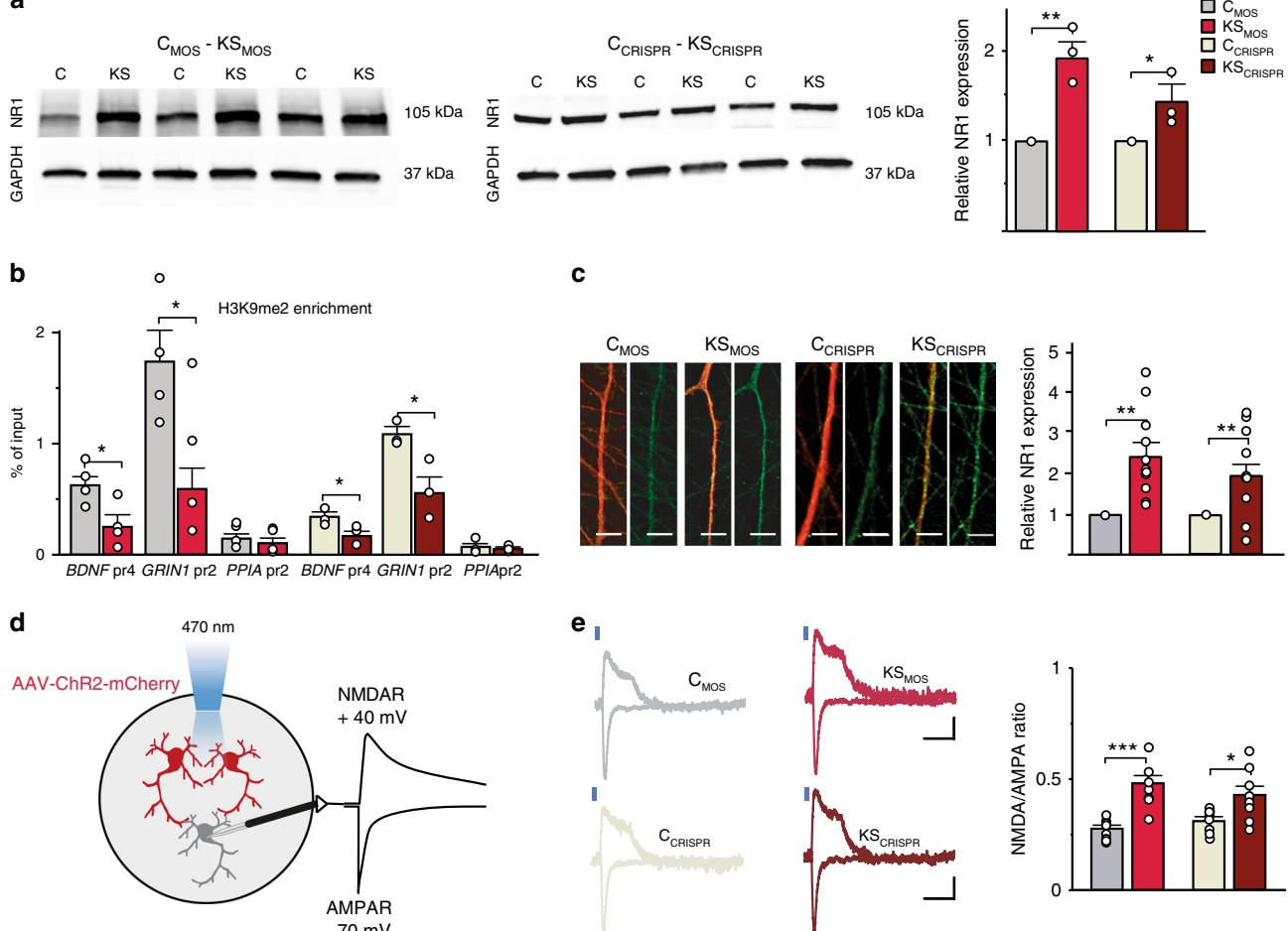

**Fig. 5** NMDA receptor subunit 1 is upregulated in KS patient-derived neurons. **a** Representative western blots and graph showing the NR1 protein levels in control and KS iNeurons at DIV 28. Values of $KS_{MOS}$ are normalized to $C_{MOS}$ and values of $KS_{CRISPR}$ are normalized to $C_{CRISPR}$ ($n = 3$). **b** ChIP assay of H3K9me2 followed by promoter-specific (BDNF_pr4, GRIN1_pr2, and PPIA_pr2) qPCR performed in iNeurons from $C_{MOS}$, $KS_{MOS}$, $C_{CRISPR}$, and $KS_{CRISPR}$ ($n = 4$ for mosaic lines and $n = 3$ for crispr lines). **c** Representative images showing NR1 expression and quantification for $C_{MOS}$, $KS_{MOS}$, $C_{CRISPR}$, and $KS_{CRISPR}$ ($n = 12$). **d** Schematic representation of methodology to obtain the AMPA and NMDA ratio. **e** Representative traces and quantification of NMDA/AMPA ratio for $C_{MOS}$, $KS_{MOS}$, $C_{CRISPR}$, and $KS_{CRISPR}$ ($n = 8$). Blue bars indicate light stimulus. Scale bar 100 ms, 10 pA. Data represent means ± SEM. *$P < 0.05$, **$P < 0.005$, ***$P < 0.0005$, one-way ANOVA test and post hoc Bonferroni correction was performed between conditions. Source data is available as a Source Data file

indicative of a shift towards AMPAR-driven networks. A discriminant analysis based on the aforementioned parameters showed that the $KS_{MOS}$ networks treated with MK-801 for 7 days became segregated from untreated $KS_{MOS}$ networks and closer to $C_{MOS}$ networks (Fig. 7h). Similar effects of MK-801 treatment were also observed in $KS_{CRISPR}$ (Supplementary Fig. 8C–G). These results indicate that the aberrant activity pattern of KS patient-derived neuronal networks can be rescued by specifically inhibiting NMDAR activity.

## Discussion
In this study, we developed a human model of KS that enabled us to identify specific functional aberrations, from gene expression to neuronal network behavior, due to *EHMT1* haploinsufficiency. We found that excitatory networks derived from KS patients showed a distinct and robust neuronal network phenotype with striking similarity across different types of mutations in *EHMT1*. The phenotype was characterized by network bursts with a longer duration, lower frequency and more irregular pattern compared to controls. At the cellular level, we demonstrated that the network phenotype was mediated by NR1 upregulation.

Interestingly, the neuronal phenotype was consistent across species and model systems. Indeed, we found that network bursts also occurred with a longer duration and irregular pattern in dissociated neuronal networks from either embryonic rats (i.e., where *Ehmt1* was downregulated through RNA interference[29,37]) or *Ehmt1*[+/−] mice. This indicates that some network parameters are consistently and similarly altered in divergent KS models. The appearance of a consistent phenotype both in a system where excitatory and inhibitory transmission were present and in a system where inhibition was absent (human iNeurons) indicates a major contribution from aberrant excitatory neurotransmission to KS pathobiology.

One major characteristic of KS networks was the longer duration of the network bursts. A change in burst length can be indicative of synaptic changes in GABARs, NMDARs, and/or AMPARs[31]. Given that inhibition was absent in our human model, we focused our analysis on NMDARs and AMPARs. We present several lines of evidence that suggest the long bursts exhibited by KS networks are mediated by upregulation of NR1. First, by acutely blocking AMPARs and NMDARs with chemical inhibitors, we show that network burst activity in KS networks is strongly dependent on NMDAR-mediated transmission, in

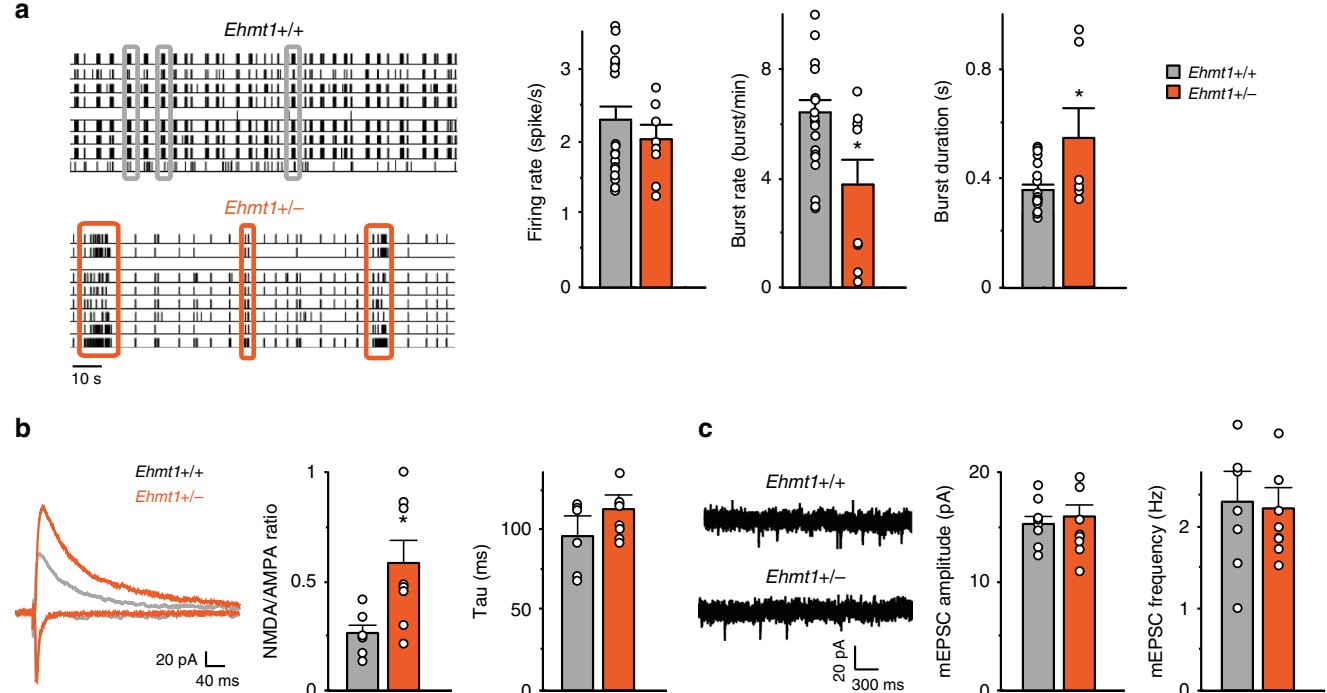

**Fig. 6** Neuronal network activity and NMDAR/AMPAR ratio in $Ehmt1^{+/-}$ mice. **a** Representative raster plot showing 2 min of recording of the activity exhibited by cultured primary neurons derived from $Ehmt1^{+/+}$ and $Ehmt1^{+/-}$ mice grown on MEAs at DIV 20. Three network burst are highlighted for each raster. Graph showing the firing rate, network bursting rate and network burst duration of cultures from $Ehmt1^{+/+}$ and $Ehmt1^{+/-}$ mice (gray and orange bar respectively) on MEA at DIV 28. $n = 8$ for $Ehmt1^{+/+}$ and $n = 6$ for $Ehmt1^{+/-}$. **b** NMDAR/AMPAR ratio and decay constant (i.e., Tau) in cortical slices (layer 2/3 auditory cortex) from $Ehmt1^{+/+}$ and $Ehmt1^{+/-}$ mice at P21. $n = 6$ for $Ehmt1^{+/+}$ and $n = 8$ for $Ehmt1^{+/-}$. **c** Representative example traces of miniature excitatory postsynaptic currents (mEPSCs) excitatory events recorded from layer 2/3 from $Ehmt1^{+/+}$ and $Ehmt1^{+/-}$ mice. Graph showing quantification of the frequency and amplitude of mEPSCs of $Ehmt1^{+/+}$ and $Ehmt1^{+/-}$ mice. $n = 8$ for $Ehmt1^{+/+}$ and $n = 8$ for $Ehmt1^{+/-}$. Data represent means ± SEM. *$P <$ 0.05, Mann–Whitney test was performed between two groups. Source data is available as a Source Data file

contrast to control networks, where network bursts are mainly dependent on AMPAR-mediated transmission. Second, we were able to reverse KS network phenotypes, including the long NBD, by blocking NMDAR activity. Third, we found that NR1 is upregulated in KS iNeurons at both, mRNA and protein level. Fourth, NR1 upregulation is paralleled by H3K9me2 hypo-methylation at the *GRIN1* promoter. Fifth, NMDAR/AMPAR ratio was increased in KS iNeurons. Finally, we found increased NMDAR-mediated currents in the cortex of $Ehmt1^{+/-}$ mice. This cross-species comparison further validates that observed effects are due to decreased EHMT1 enzymatic function and show that development of network activity in human and mouse cortical networks may follow evolutionarily conserved and stable epige-netic programming.

Genetic evidence has directly implicated NMDARs in NDDs. For example, multiple heterozygous mutations in NMDAR sub-unit genes have been identified to be causal for ID, ASD, or epilepsy[38]. NMDAR dysfunction is mainly attributed to hypo-function, but there are observations associating NMDAR hyper-function to ID/ASD:[39] upregulated NR1 protein levels were found in the cerebellum of ASD patients[40] and the NR2A and NR2B subunits of the NMDAR were found to be upregulated in the valproic acid animal model of autism[41]. In the Rett syndrome mouse model[42], loss of *MECP2* function resulted in develop-mental dysregulation of NMDAR expression[35,43]. Of note, work in mouse models of NDDs show that the changes in NMDARs expression are temporally and spatially restricted. This illustrates the importance of evaluating at what developmental age and in which brain regions changes in NMDAR expression occur, which is especially relevant for the design of rescue strategies[44–48]. Our

data shows that NR1 is upregulated in KS iNeurons of cortical identity and support the idea that dysfunctional glutamatergic neurotransmission plays a role in NDDs.

By blocking NMDARs in KS networks, we were able to induce the early phase of synaptic upscaling enabling the incorporation of GluA2-lacking receptors, followed by the insertion of GluA2-containing receptors. This plasticity mediated by AMPARs was more easily initiated in KS than in control networks and allowed KS networks to switch, at least temporarily, from a mainly NMDAR-driven network to an AMPAR-driven network. Inter-estingly, we also observed that Naspm had an effect on control cells that were treated with NMDAR antagonists, suggesting that GluA2-lacking receptor were inserted and/or possibly exchanged for GluA2-containing receptors after NMDAR antagonist treat-ment. Together, this suggests that loss of EHMT1 could facilitate NMDAR-mediated plasticity after a comparatively milder sti-mulus, in this case NMDAR blockade with D-AP5. This is in agreement with a recent study showing that inhibition of EHMT1/EHMT2 activity reinforces early LTP in an NMDAR-dependent manner[18]. The authors showed that pharmacologically blocking EHMT1/2 before a mild LTP stimulus increased the LTP response, highlighting a role for EHMT1 and associated H3K9me2 in metaplasticity[18]. A link between NMDARs and EHMT1 has also been shown in vivo where NMDAR activity regulates the recruitment of EHMT1/2 and subsequent H3K9me2 levels at target gene promoters in the lateral amygdala, in the context of fear learning[49]. This, together with our data, suggests that there is a reciprocal interaction, between NMDAR activity and EHMT1 function, and a positive-feedback mechanism where EHMT1 methylates the NR1 promoter upon activation by

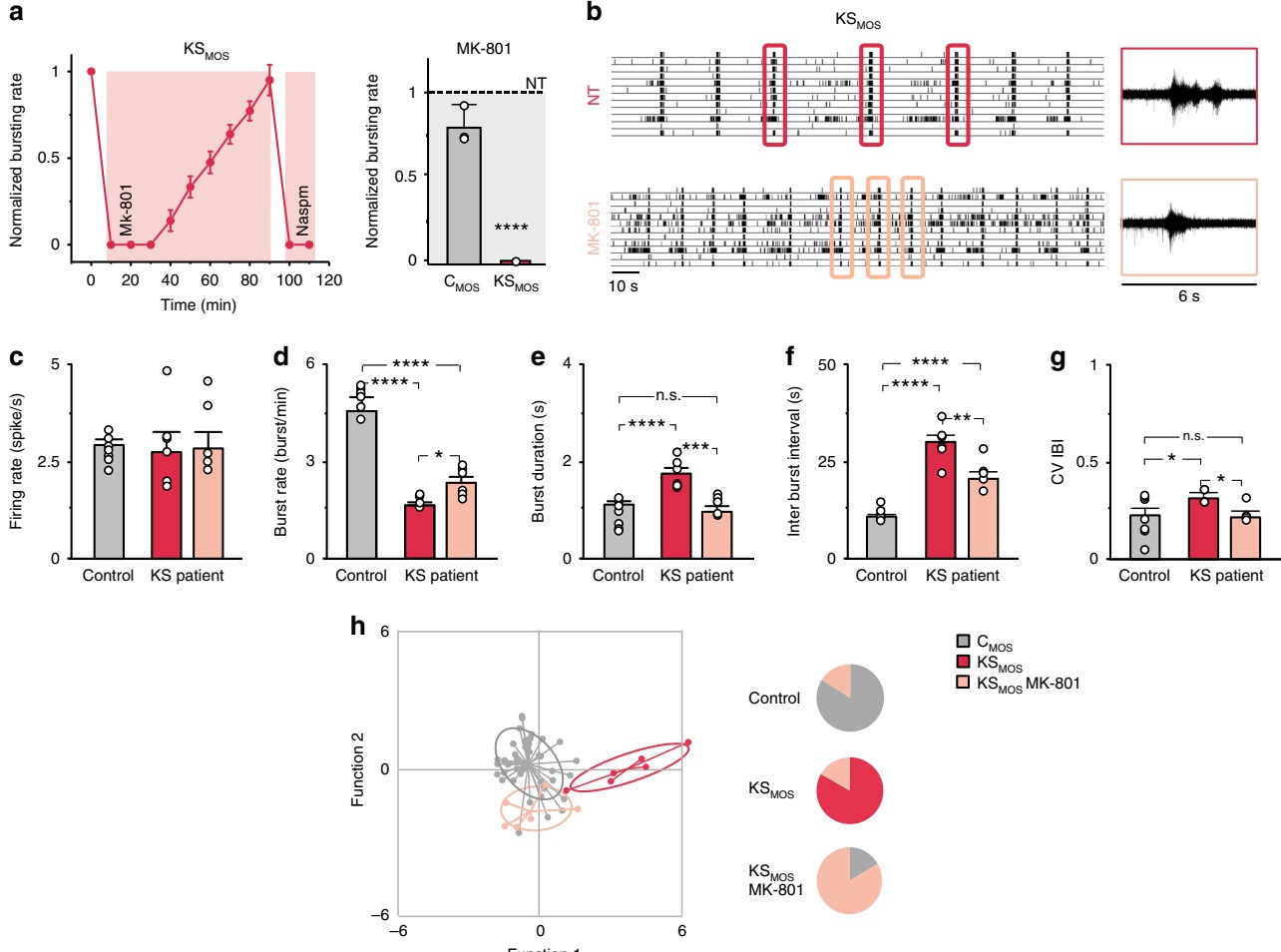

**Fig. 7** NMDAR antagonist MK-801 rescues KS network phenotypes. **a** MK-801 (1 μM) effect on KS$_{MOS}$ neuronal network activity (DIV 28). After 90 min, Naspm was added. Graph showing the effect of MK-801 (1 μM) treatment on the neuronal network burst frequency for C$_{MOS}$ and KS$_{MOS}$ derived neuronal network 20 min after application. The values are normalized to the nontreated (NT) condition. **b** Representative raster plot showing the activity exhibited by KS patient-derived neuronal network (KS$_{MOS}$) grown on MEAs either nontreated or after one week of treatment with MK-801. Six second of raw data showing a burst recorded from a representative channel are shown for the NT and the treated conditions. **c**–**g** Quantification of network properties as indicated, $n = 8$ for C$_{MOS}$; $n = 6$ for KS$_{MOS}$ and $n = 6$ for KS$_{MOS}$ treated with MK-801. **h** Canonical scores plots based on discriminant analyses for control (i.e., C$_1$, C$_2$, C$_{MOS}$, and C$_{CRISPR}$), KS$_{MOS}$ and KS$_{MOS}$ treated with MK-801 (84% correct classification). Discriminant functions are based on using the following network activity parameters: firing rate, network burst rate, network burst duration, percentage of spike outside network burst and coefficient of variability of the inter-burst interval. Group envelopes (ellipses) are centered on the group centroids, $n = 50$ for controls; $n = 6$ for KS$_{MOS}$ and KS$_{MOS}$ treated with MK-801. Pie charts visualize accuracy of discriminant analyses functions by showing the relative distribution of lines per a-priory group after reverse testing for group identity. Data represent means ± SEM. *$P < 0.05$, ***$P < 0.0005$, ****$P < 0.0001$, one-way ANOVA test and post hoc Bonferroni correction was performed between conditions. Source data is available as a Source Data file

NMDARs. If and under which circumstances such a feedback mechanism is active during development and/or in adulthood deserves further investigation.

At the molecular level we and others have shown that EHMT1 haploinsufficiency can cause modest transcriptional changes[12,16,37,50]. We found that the upregulation of NR1 correlates with reduced H3K9me2 occupancy at the *GRIN1* promoter. This suggests that during normal development EHMT1 is directly involved in the regulation of NR1 expression. However, we cannot exclude that EHMT1 also regulates NR1 expression via other, less direct mechanisms. For example, EHMT1 is a member of a large complex that includes the neuron-restrictive silencer factor (NRSF/REST) repressive unit, which is important for repressing neuronal genes in progenitors, including NR1[51–53]. In addition, growth factors such as BDNF, which is increased upon EHMT1 deletion[16], have been shown to increase NR1 mRNA levels in cultured embryonic cortical neurons[54]. Finally, we know

from previous studies that EHMT1 regulates genome-wide H3K9me2 deposition, altering the expression of multiple genes[12,16,37,50].

An important aspect of our study is the identification of a robust and consistent network phenotype linked to KS on MEAs. We show that KS networks differed from controls based on a set of parameters describing the neuronal network activity and using discriminant analysis. Our analysis showed that individual unrelated controls clustered with little variation, which was exemplified by the fact our predictive group membership analysis was able to accurately assign control networks to the control group (Fig. S6). In contrast, we found that KS patient lines significantly differed from controls, including their respective isogenic controls (C$_{MOS}$ vs. K$_{MOS}$ and C$_{CRISPR}$ vs. KS$_{CRISPR}$). It should be noticed here that the 233 kB deletion in KS$_{MOS}$ incorporates another gene (*CACNA1B*) that might affect neuronal function, which might explain minor changes compared to the

other KS lines. When we performed predictive group membership analysis, we found that individual KS networks were mostly assigned to the corresponding patient line (Fig. S6), indicating that this level of analysis on neuronal networks can potentially detect patient-specific phenotypic variance that arises early in development. It is foreseeable that, in future studies, an in-depth interrogation of the network activity for networks consisting of different human derived cell types or networks from brain organoids would allow measuring more complex neuronal signals on MEAs. This would be especially relevant for the stratification of genetically complex disorders (e.g., idiopathic forms of ASD) as these in vitro network phenotypes could then be used as endo-phenotype in pharmacological studies.

We show that it is possible to rescue the neuronal phenotype by blocking NMDARs in mature networks, a finding that has important clinical relevance. For example, NMDAR antagonists like ketamine and memantine have been used successfully in mouse models to treat RTT[46,47] and other NDDs[45,48] and led in some cases to improvements in small open-label trials for autism[55–57]. These studies, as well as ours, provide preclinical proof of concept that NMDAR antagonists could ameliorate neurological dysfunction and reverse at least some circuit-level defects. Furthermore, our observation that neuronal phenotypes can be rescued in mature networks agrees with previous data showing that reinstating EHMT1 function in adult flies is sufficient to rescue memory deficits[13]. This adds to a growing list of genetically defined ID syndromes that might be amenable to postnatal therapeutic intervention[58].

Summarized, our study shows that combining iPS cell-derived human neuronal models with neuronal network dynamics is a promising tool to identify novel targets for possible treatment strategies for NDDs, such as ID and ASD.

## Methods

**Patient information and iPS cell generation**. In this study we used in total four control and four iPS cells with reduced EHMT1 function. In contrast to a previous study[59] we included patients in this study that present the full spectrum of KS associated symptoms, including ID and ASD. $KS_1$ and $KS_2$ originate from two individuals: a female KS patient with a frameshift and a missense mutation in the EHMT1 gene, respectively (patient 25 in ref. [22] and patient 20 in ref. [8]). An isogenic pair of control ($C_{MOS}$) and KS line ($KS_{MOS}$) originated from an individual diagnosed with a mosaic heterozygous 233 kbp deletion of chromosome 9 which includes the EHMT1 gene (deletion starts after exon 4) and CACNA1B gene[23]. Clones were selected after performing RT-qPCR on genomic DNA with primers spanning Exon 3: F-GAAGCAAAACCACGTCACTG; R-GTAGTCCTCAAGGG CTGTGC. Exon 4: F-CCCAGAGAAGTTCGAGAAGC, R- GGGTAAAAGCTG CTGTCGTC. Exon 5 F-CAGCTGCAGTATCTCGGAAG, R- AACATCTCAATC ACCGTCCTC. Exon 6 F- GACTCGGATGAGGACGACTC, R- GGAAGTCCTGC TGTCCTCTG. All iPS cells used in this study were obtained from reprogrammed fibroblasts. For most of the lines reprogramming to iPS cells was induced by retroviral vectors expressing four transcription factors: Oct4, Sox2, Klf4, and cMyc. For control line ($C_2$), which was obtained from Mandegar et al.[60], episomal reprogramming was performed, whereas the $C_{CRISPR}$ line was generated by non-integrating Sendai virus. Generated clones (at least two per patient line) were selected and tested for pluripotency and genomic integrity based on SNP arrays. IPS cells were always cultured on Matrigel (Corning, #356237) in E8 flex (Thermo Fisher Scientific) supplemented with primocin (0.1 μg/ml, Invivogen) and low puromycin (0.5 μg/ml) and G418 concentrations (50 μg/ml) at 37 °C/5% $CO_2$. Medium was refreshed every 2–3 days and cells were passaged twice per week using an enzyme-free reagent (ReLeSR, Stem Cell Technologies). Collecting patient material and establishing hiPSCs have all been performed according to locally (Radboudumc) IRB protocols.

**CRISPR/Cas9 editing of EHMT1**. We made use of the CRISPR/Cas9 technology in order to create a heterozygous EHMT1 mutation in Exon 2 in a iPS cell line derived from a healthy 51-year-old male to mimic KS in isogenic cell lines, generated by KULSTEM (Leuven, Belgium). In brief, $1 \times 10^6$ iPSCs in single-cell suspension were nucleofected with Cas9 ribonucleoprotein complexes (10 μg/reaction S.p. Cas9 Nuclease 3NLS (IDT 1074181), CRIPSR crRNAs (IDT) and CRISPR tracrRNA (IDT 1072532) 0.4 nmoles each/reaction) and a donor vector (5 μg/reaction). The two crRNAs were designed to specifically target EHMT1 (TCTAACAGGCAGTTCCGGCGAGG and TAACAGGCAGTTCCGGCGA

GGGG). The donor vector was a piggyback construct containing a hygromycin selection cassette as well as sequences that enable homology-directed repair ensuring the insertion of premature stop codons in Exon 2 of the EHMT1 gene. Cells were nucleofected using the Human Stem Cell Nucleofector® Kit 2 (Lonza, VPH-5022) in combination with the AMAXA-2b nucleofector, program F16. After nucleofection cells were resuspended in E8 flex (Thermo Fisher Scientific) supplemented with Revitacell (Thermo Fisher Scientific). When the iPS cells reached a confluency of about 40% selection was started using 5 μg/ml hygromycin. The antibiotics concentration was increased to up to 200 μg/ml over 2 weeks. Hygromycin resistant colonies were picked and PCR validation was used to ensure heterozygous editing of Exon2. To prove reduced EHMT1 expression in the CRISPR/Cas9 edited clone RT-qPCR and Western Blot were performed to measure EHMT1 protein levels in $C_{Crispr}$ and $KS_{Crispr}$. DNA fingerprinting profile was performed on both lines by qPCR detection of a reference set of SNP panel using the TaqMan assays from Life Technologies. The genome-edited iPS cell line shows identical SNP profile with the corresponding iPS cell line used for gene targeting. Off-target of genome editing was verified by sequencing the top four off-target sites of each gRNA have been sequenced and no INDELs were detected (Supplementary Fig. 5). The generated iPS cells were validated for pluripotency markers and quantitative analysis of tri-lineage differentiation potential was performed. All generated iPS cell lines have the capacity to differentiate toward all three germ layers (endoderm, mesoderm, ectoderm). To this end embryoid bodies were generated in 24-well Corning low attachment plates. For spontaneous differentiation, the culture was kept for 7 days in E6 medium (Thermo Fisher Scientific). The medium was changed every 2 days. Cells were harvested after 7 days for RNA extraction with the GenElute Mammalian Total RNA kit (Sigma). cDNA synthesis was performed with Superscript III and used for qPCR according to manufacturer's protocol with TaqMan human iPS cell Scorecard assay (Life Technologies). Data analysis was performed with Scorecard software (online tool Life technologies), comparing with a reference set of pluripotent stem cell lines.

**Western Blot**. For Western Blot cell lysates were made from confluent wells in six-well plates of either iPS cells or iNeurons. Medium was always refreshed 4 h beforehand. Protein samples were loaded, separated by sodium dodecyl supfate (SDS) polyacrylamide gel electrophoresis, and transferred to nitrocellulose membrane (BIO-RAD). The membrane was then probed with an EHMT1 antibody (1:1000; Abcam ab41969) or NMDAR1 (1:100; Biolegend 818601). To control for loading differences, we probed with anti-gamma tubulin (1:1000; Sigma T5326) or GAPDH (1:1000; Cell Signaling #2118). For visualization horseradish peroxidase-conjugated secondary antibodies were used (1:20000 for both; goat anti-mouse: Jackson ImmunoResearch Laboratries, 115-035-062. Goat-anti rabbit: Invitrogen, A21245).

**Neuronal differentiation**. iPS cells were directly derived into, excitatory cortical Layer 2/3 neurons by overexpressing the neuronal determinant Neurogenin 2 (NGN2) upon doxycycline treatment based on Zhang et al.[24] and as we described previously[25]. To support neuronal maturation, freshly prepared rat astrocytes[25] were added to the culture in a 1:1 ratio two days after plating. At DIV 3 the medium was changed to Neurobasal medium (Thermo Fisher Scientific) supplemented with B-27 (Thermo Fisher Scientific), glutaMAX (Thermo Fisher Scientific), primocin (0.1 μg/ml), NT3 (10 ng/ml), BDNF (10 ng/ml), and doxycycline (4 μg/ml). Cytosine b-D-arabinofuranoside (Ara-C) (2 μM; Sigma) was added once to remove any proliferating cell from the culture. From DIV 6 onwards half of the medium was refreshed three times a week. The medium was additionally supplemented with 2.5% FBS (Sigma) to support astrocyte viability from DIV 10 onwards. Neuronal cultures were kept through the whole differentiation process at 37 °C/5% $CO_2$.

**Cortical cultures from mice**. Primary cortical neurons were prepared from $Ehmt1^{+/-}$ and $Ehmt1^{+/+}$ mice from individual E16.5 embryos as previously described[16]. Since the genotype was unknown at the time of harvest, each embryo was collected and the brains were processed separately. Each whole brain was kept on ice in 1 mL L-15 medium, organized separately in a 24-well plate, and tail clips were collected for genotyping.

**Neuronal morphology reconstruction**. To examine morphology of neurons cells on coverslips were transfected with plasmids expressing Discosoma species red (dsRED) fluorescent protein one week after plating. DNAin (MTI-GlobalStem) was used according to manual instructions. Medium was refreshed completely the day after DNAin application. After the treatment cells were cultured as previously described.

After 3 weeks of differentiation cells were fixed in 4% paraformaldehyde/4% sucrose (v/v) in phosphate-buffered saline (PBS) and mounted with DAKO. Transfected neurons were imaged using a Zeiss Axio Imager Z1 and digitally reconstructed using Neurolucida 360 software (MBF–Bioscience, Williston, ND, USA). For large cells multiple 2-dimensional images of these neurons were taken and subsequently stitched together using the stitching plugin of FIJI 2017 software. The three-dimensional reconstructions and quantitative morphometrical analyses focused on the somatodendritic organization of the neurons. We defined origins for

individual primary dendrites by identifying emerging neurites with diameters that were less than 50% of the diameter of the soma. Axons, which were excluded from reconstructions and further analysis, we visually identified by their long, thin properties, far reaching projections and numerous directional changes Neurons that had at least two primary dendrites and reached at least the second branching order were considered for analysis. For morphometrical analysis we determined soma size, number of primary dendrites, length and branching points per primary dendrite and total dendritic length. To measure the total span of the dendritic field (receptive area) of a neuron we performed convex hull 3D analysis. Note, that due to the two-dimensional nature of the imaging data, we collapsed the convex hull 3D data to two-dimensions, resulting in a measurement of the receptive area and not the volume of the span of the dendritic field. Furthermore, Sholl analysis was performed to investigate dendritic complexity in dependence form distance to soma. For each distance interval (10 μm each) the number of intersections (the number of dendrites that cross each concentric circle), number of nodes and total dendritic length was measured. Discriminant function analysis with canonical discriminant functions and reclassification of group membership based on parameters describing neuronal morphology were performed in SPSS.

**Immunocytochemistry**. Cells were fixed with 4% paraformaldehyde/4% sucrose (v/v) for 15 min and permeabilized with 0.2% triton in PBS for 10 min at RT. Nonspecific binding sites were blocked by incubation in blocking buffer (PBS, 5% normal goat serum, 1% bovine serum albumin, 0.2% Triton) for 1 h at RT. Primary antibodies were diluted in blocking buffer incubated overnight at 4 °C. Secondary antibodies, conjugated to Alexa-fluorochromes, were also diluted in blocking buffer and applied for 1 h at RT. Hoechst 33342 (Molecular Probes) was used to stain the nucleus before cells were mounted using DAKO fluorescent mounting medium (DAKO). Neurons were fixed at DIV 21 for synaptic stainings. Used primary antibodies were: mouse anti-MAP2 (1:1000; Sigma M4403); guinea pig anti-MAP2 (1:1000; Synaptic Systems 188004); guinea pig anti-synapsin 1/2 (1:1000; Synaptic Systems 106004); mouse anti-PSD95 (1:50; Thermo Fisher Scientific MA1-045); rabbit anti-GFAP (1:500; Abcam ab7260), mouse anti-pan axon (1:1000; Covance SMI-312R), mouse anti-NMDAR1 (1:1000; ThermoFisher Scientific 54.1). Secondary antibodies that were used are: goat anti-guinea pig Alexa Fluor 647 (1:2000, Invitrogen A-21450); goat anti-rabbit Alexa Fluor 488 (1:2000, Invitrogen A-11034); goat anti-rabbit Alexa Fluor 568 (1:2000, Invitrogen A-11036); goat anti-mouse Alexa Fluor 488 (1:2000, Invitrogen A-11029); goat anti-mouse Alexa Fluor 568 (1:2000, Invitrogen A-11031). We imaged at a 63× magnification using the Zeiss Axio Imager Z1 equipped with apotome. All conditions within a batch were acquired with the same settings in order to compare signal intensities between different experimental conditions. Signals were quantified using FIJI software. The number of synaptic puncta was determined per individual cell via manual counting and divided by the dendritic length of the neuron.

To investigate H3K9me2 iNeurons were fixed at DIV3. Used primary antibodies are: rabbit anti-H3K9me2 antibody (1:500, Millipore 07-441), and guinea pig anti-MAP2 antibody (1:1000, Synaptic Systems 188004). Secondary antibodies used are: goat anti-rabbit Alexa Fluor 488 (1:2000, Invitrogen A-11034) and goat anti-guinea pig Alexa Fluor 647 (1:2000, Invitrogen A-21450). Epifluorescent images were taken with the same exposure time using Zeiss Axio Imager Z1 with apotome. H3K9me2 was assessed based on color intensity measurements by ImageJ analysis tools.

**Quantification of mRNA by RT-qPCR**. RNA samples were isolated from both, iPS cells or iNeurons (DIV28) using Nucleo Spin RNA isolation kit (Machery Nagel, 740955.250) according to the manufacturer's instructions. RNA samples (200 ng) were converted into cDNA by iScript cDNA synthesis kit (BIO-RAD, 1708891). cDNA products were cleaned up using the Nucleospin Gel and PCR clean-up kit (Machery Nagel, 740609.250). Human-specific primers were designed with Primer3plus and IDT PrimerQuest tools, respectively. Primer sequences are given in supplementary Table 1. qPCRs were performed in the 7500 Fast Real Time PCR System apparatus (Applied Biosystems) with GoTaq qPCR master mix 2× with SYBR Green (Promega, A600A) according to the manufacturer's protocol. The PCR program was designed as following: After an initial denaturation step at 95 °C for 10 min, PCR amplifications proceeded for 40 cycles of 95 °C for 15 s and 60 °C for 30 s and followed by a melting curve. All samples were analyzed in duplicate in the same run, placed in adjacent wells. Reverse transcriptase-negative controls and no template-controls were included in all runs. The arithmetic mean of the Ct values of the technical replicas was used for calculations. Relative mRNA expression levels were calculated using the $2^{-\Delta\Delta Ct}$ method[61] with standardization to housekeeping genes.

**Chromatin immunoprecipitation (ChIP)-qPCR**. ChIP was performed according to the Blueprint-IHEC (international human epigenome consortium) protocol with slight modifications. Briefly, 5,00,000 cells were used for one immunoprecipitation reaction. Cells were first crosslinked with disuccinimidyl glutarate at a final concentration of 2 mM for 30 min at RT followed by second crosslinking treatment with 1% formaldehyde for 8 min at RT. Crosslinking was stopped by adding glycine to a final concentration of 0.125 M for 10 min. Samples were washed and homogenized in cold lysis buffer (10 mM Tris (pH 8.0), 0.2%NP40, 0.2% TritonX-100, 1

mM EDTA, 0.5 mM EGTA, 5× protease inhibitors EDTA-free, and 10 mM NaCl) for 15 min on ice. Nuclei were lysed with nuclei lysis buffer (1 mM EDTA, 0.5 mM EGTA, 50 mM HEPES) for 20 min rotating at 4 °C. After centrifugation the collected extracts were resuspended in sonication buffer (20 mM HEPES pH = 7.6, 1% SDS, 2.5× protease inhibitors EDTA-free) and sonicated with Bioruptor Power-up (Diagenode) to shear DNA into 150–700 bp fragments. Cleaned-up chromatin was diluted with ChIP dilution buffer (0.15% SDS, 1% TritonX-100, 1.2 mM EDTA, 16.7 mM Tris pH = 8, 167 mM NaCl). Immunoprecipitation was performed with anti-H3K9me2 (5 μg/reaction, Abcam ab1220) and anti-EHMT1 (15 μg/reaction, Abcam ab41969) antibodies overnight rotating at 4 °C. As a negative control, anti-human IgG (10 μg/reaction, Abcam ab2410) was used in place of specific antibodies. Chromatin was enriched by 10 μL of each Protein A (Invitrogen, 10001D) and Protein G Dynabeads (Invitrogen, 10003D) for 2 h rotating at 4 °C. Beads were washed once with wash buffer 1 (2 mM EDTA, 20 mM Tris pH = 8, 1% TritonX-100, 0.1% SDS, 150 mM NaCl), twice with wash buffer 2 (2 mM EDTA, 20 mM Tris pH = 8, 1% TritonX-100, 0.1% SDS, 500 mM NaCl), twice with wash buffer 3 (1 mM EDTA, 10 mM Tris pH = 8) and eluted in elution buffer (1% SDS; 100 mM NaHCO3) by rotating for 20 min at RT. Eluate was de-crosslinked in elution buffer with 40 μg proteinase-K and 200 mM NaCl overnight at 65 °C. DNA material was cleaned with QIAquick MinElute PCR Purification Kit (Qiagen, 28006) according to manufacturer's instructions. Five microliterof eluted DNA sample was used for qPCR analysis. qPCRs were performed as described earlier. For primer sequences see Supplementary Table 2. All samples were analyzed in duplicate in the same run and placed in adjacent wells. ChIP-qPCR results are expressed as % input, which was calculated by the following formula: $100 \times 2^{-\Delta Ct}$, where $\Delta Ct = Ct[ChIP] - (Ct [input] - Log_2(input dilution factor))$.

**MEA recordings and data analysis**. All recordings were performed using the 24-well MEA system (Multichannel Systems, MCS GmbH, Reutlingen, Germany). MEA devices are composed by 24 independent wells with embedded microelectrodes (i.e., 12 electrodes/well, 80 μm in diameter and spaced 300 μm apart). Spontaneous electrophysiological activity of iPS cell-derived neuronal network grown on MEAs was recorded for 20 min. During the recording, the temperature was maintained constant at 37 °C, and the evaporation and pH changes of the medium was prevented by inflating a constant, slow flow of humidified gas (5% $CO_2$ and 95% $O_2$) onto the MEA plate (with lid on). The signal was sampled at 10 kHz, filtered with a high-pass filter (i.e., Butterworth, 100 Hz cutoff frequency) and the noise threshold was set at ±4.5 standard deviations.

Data analysis was performed off-line by using Multiwell Analyzer (i.e., software from the 24-well MEA system that allows the extraction of the spike trains) and a custom software package named SPYCODE developed in MATLAB (The Mathworks, Natick, MA, USA) that allows the extraction of parameters describing the network activity[62].

The mean firing rate (MFR) of the network was obtained by computing the firing rate of each channel averaged among all the active electrodes of the MEA. Burst detection: bursts were detected using a Burst Detection algorithm. The algorithm is based on the computation of the logarithmic inter-spike interval histogram in which inter-burst activity (i.e., between bursts and/or outside bursts) and intra-burst activity (i.e., within burst) for each recording channel can be easily identified, and then a threshold for detecting spikes belonging to the same burst is automatically defined. From the burst detection, the number of bursting channels (above threshold 0.4 burst/s and at least 5 spikes in burst with a minimal inter-spike-interval of 100 ms) was determined. Network burst detection: synchronous events were detected by looking for sequences of closely spaced single-channels bursts. A network burst was defined as burst that occurs in >80% of the network active channels. The distributions of NBD's and network inter-burst interval (NIBI, interval between two consecutive network bursts) were computed using bins of 100 ms and 1 s respectively. Network burst irregularity: irregularity was estimated by computing the CV of the NIBI, which is the standard deviation divided by the mean of the NIBI. Discriminant function analysis with canonical discriminant functions and reclassification of group membership based on parameters describing neuronal network activity were performed in SPSS.

**Chemicals**. All reagents were prepared fresh into concentrated stocks as indicated below, and stored frozen at −20 °C. The following compounds were used in pharmacological experiments: 1-Naphthyl acetyl spermine trihydrochloride ("Naspm", 100 mM in PBS, Tocris Cat No 2766); (+)-MK 801 maleate ("MK-801", 100 mM in DMSO, Tocris Cat No 0924); 2,3-Dioxo-6-nitro-1,2,3,4-tetra-hydrobenzo[f]quinoxaline-7-sulfonamide ("NBQX", 100 mM in DMSO, Tocris Cat No 0373); D-2-amino-5-phosphonovalerate ("D-AP5", 50 mM in PBS, Tocris Cat No 0106); Retigabine (100 mM in DMSO, Tocris Cat No. 6233). For all experiments on MEAs, and immediately before adding a compound to the cells, an aliquot of the concentrated stock was first diluted 1:10 in room temperature DPBS and vortexed briefly. Then, the appropriate amount of working dilution was added directly to wells on the MEA and mixing was primarily through diffusion into the (500 μl) cell culture medium. Of note, where relevant the DMSO concentration in the medium was always ≤0.05% v/v.

**Pharmacological experiments**. Control and KS patient neuronal networks were treated acutely with D-AP5 (60 µM), NBQX (50 µM), MK-801 (1 µM), Naspm (10 µM), or Retigabine (10 µM) at DIV 28 after a 20-min recording of spontaneous activity. Then, the recording was paused, the compounds were added to the MEA, and the recording was restarted after 5 min. We recorded neuronal network activity for 60 min with D-AP5 or NBQX, 90 min with MK-801, 100 min with Retigabine, and for 20 min after the addition of Naspm. In experiments where we examined the effect of chronic NMDAR blockade, control and KS patient neuronal networks were treated with 1 µM MK-801 starting at DIV 28 and lasting 7 days total. MK-801 was replenished every two days, where it was freshly diluted to 1 µM in complete Neurobasal medium for the routine half-medium change. All experiments were performed at 37 °C.

**Single-cell electrophysiology**. For single-cell recordings we used neurons derived from $C_1$, $C_2$, $C_{MOS}$, $C_{CRISPR}$, and $KS_1$, $KS_2$, $KS_{MOS}$, and $KS_{CRISPR}$ after 3 weeks of differentiation. Experiments were performed in a recording chamber on the stage of an Olympus BX51WI upright microscope (Olympus Life Science, PA, USA) equipped with infrared differential interference contrast optics, an Olympus LUMPlanFL N 40× water-immersion objective (Olympus Life Science, PA, USA) and kappa MXC 200 camera system (Kappa optronics GmbH, Gleichen, Germany) for visualization. Recordings were made using a SEC-05× amplifier (NPI Electronic GmbH, Tamm, Germany), low-pass filtered at 3 kHz and digitized at 10 kHz using a 1401 acquisition board and recorded with Signal (CED, Cambridge, UK). Recordings were not corrected for liquid junction potential (±10 mV).

We performed the recordings of neurons cultured on cover slips under continuous perfusion with oxygenated (95% $O_2$/5% $CO_2$) artificial cerebrospinal fluid (ACSF) at 30 °C containing (in mM) 124 NaCl, 1.25 $NaH_2PO_4$, 3 KCl, 26 $NaHCO_3$, 11 Glucose, 2 $CaCl_2$, 1 $MgCl_2$ (adjusted to pH 7.4). Patch pipettes (6–8 MΩ) were pulled from borosilicate glass with filament and fire-polished ends (Science Products GmbH, Hofheim, Germany) using the PMP-102 micropipette puller (MicroData Instrument, NJ, USA). For current clamp recordings of the intrinsic electrophysiological properties, we filled pipettes with a potassium-based solution containing 130 mM K-Gluconate, 5 mM KCl, 10 mM HEPES, 2.5 mM $MgCl_2$, 4 mM Na2-ATP, 0.4 mM Na3-ATP, 10 mM Na-phosphocreatine, 0.6 mM EGTA (adjusted to pH 7.25 and osmolarity 290 mosmol). We determined the resting membrane potential ($V_{rmp}$) after achieving whole-cell configuration and only considered neurons with $V_{rmp}$ of −55 mV and lower for further testing and analyses. Recordings were not corrected for liquid junction potentials of ca. −10 mV. All further electrophysiological characterizations we performed at a holding potential of −60 mV. We determined the passive membrane properties via a 0.5 s hyperpolarizing current of −25 pA. Active intrinsic properties, i.e., action potential (AP) firing characteristics we derived from the first AP elicited by a 0.5 s depolarizing current injection just sufficient to reach AP threshold. For voltage clamp recordings we filled pipettes with a cesium-based solution containing 115 mM cesium methanesulfonate, 20 mM CsCl, 10 mM HEPES, 2.5 mM $MgCl_2$, 4 mM Na2ATP, 0.4 mM Na3GTP, 10 mM sodium phosphocreatine, and 0.6 mM EGTA adjusted to a pH of 7.4. Spontaneous AP-sEPSC were recorded in drug-free ACSF. Bursts were detected when at least 5 events with a minimal inter-event-interval of 100 ms were exhibited by a cell. Burst detection, burst frequency and duration were computed with software developed in MATLAB (The Mathworks, Natick, MA, USA). mEPSC were measured in voltage clamp at a holding potential of −60 mV in the presence of 1 µM TTX (Tocris, Bristol, UK) and 100 µM picrotoxin (Tocris, Bristol, UK). Evoked responses (AMPAR and NMDAR) were measured by infecting iNeurons at DIV 7 with 0.5 µl AAV-mCherry-Chr2 (UNC Vector Core, USA). Uninfected cell were recorded in voltage clamp at a holding potential of −70 mV (AMPAR) or +40 mV (NMDAR). Light-evoked synaptic responses were induced by exciting nearby infected cells (mCherry positive) with blue light (COOLLED pE-200; 10 ms, 470 nm). Stimulus strength, location of stimulus were adjusted to 30–50 pA postsynaptic AMPAR-mediated responses at −70 mV holding voltage, and 30 sweeps were averaged for the final trace. AMPA amplitudes were quantified as the peak amplitude at −70 mV holding voltage, and the NMDA amplitudes were quantified as the average of 5 ms of the trace 65 ms after the stimulus to prevent contamination of AMPAR-mediated response at +40 mV. NMDA/AMPA ratios were calculated by averaging responses in Clampfit (vs. 10.7, Molecular Devices, CA, USA). Intrinsic electrophysiological properties were analyzed using Signal and MatLab (MathWorks, MA, USA), while mEPSCs were analyzed using MiniAnalysis 6.0.2 (Synaptosoft Inc, Decatur, GA, USA) as described earlier[16]. Discriminant function analysis with canonical discriminant functions and reclassification of group membership based on neuronal intrinsic properties were performed in SPSS.

**Acute slice electrophysiology**. We used litter-matched $Ehmt1^{+/−}$ and $Ehmt1^{+/+}$ mice of adolescent age (postnatal day 21–24). Acute slices were prepared as described earlier[16]. In brief, animals were deeply anesthetized with isoflurane, then quickly decapitated. 350-µm-thick coronal slices were cut using a microtome (HM650V, Thermo Scientific) in ice-cold "cutting and storage" ACSF containing 87 mM NaCl, 11 mM glucose, 75 mM sucrose, 2.5 mM KCl, 1.25 mM $NaH_2PO_4$, 0.5 mM $CaCl_2$, 7 mM $MgCl_2$, and 26 mM $NaHCO_3$, continuously oxygenated with 95% $O_2$/5% $CO_2$. Slices were incubated for 1 h at 32 °C, after which they were

allowed to cool down to room temperature. The slices were then transferred to a recording setup as described above, and incubated in recording ACSF (124 mM NaCl, 10 mM glucose, 3 mM KCl, 1.25 mM $NaH_2PO_4$, 2 mM $CaCl_2$, 1 mM $MgCl_2$, and 26 mM $NaHCO_3$) at 30 °C with added 100 mM Picrotoxin to block GABAergic transmission. A bipolar electrode (CE2C55, FHC) coupled to an SD9 stimulator (Grass Instruments, RI, USA) was inserted into layer 4 of the auditory cortex, and pyramidal cells were patched in layer 2/3 above the bipolar electrode (<200 µm lateral distance) using 3–6 MΩ borosilicate pipettes filled with a Cesium-based intracellular solution containing 115 mM $CsMeSO_3$, 20 mM $CsCl_2$, 10 mM HEPES, 2.5 mM $MgCl_2$, 4 mM Na2ATP, 0.4 mM NaGTP, 10 mM Na-Phosphocreatine, 0.6 mM EGTA, and 5 mM QX-314. The cells were held in voltage-Clamp mode controlled by an SEC 05-LX amplifier (NPI, Tamm, Germany), low-pass filtered at 3 kHz and sampled at 20 kHz with a Power-1401 acquisition board and Signal software (CED, Cambridge, UK). Data were analyzed in Clampfit 10.7 (Molecular Devices). The AMPA trace was measured at −70 mV holding potential; stimulus strength was adjusted to 50–100 pA postsynaptic response, and 60 sweeps were averaged for the final trace. NMDA was measured at +40 mV holding potential. Cells were discarded if the measured AMPA response contained multiple peaks, indicating multisynaptic input, or if the averaged amplitude was below 25 pA. AMPA amplitudes were quantified as the peak amplitude at −70 mV holding voltage, and the NMDA amplitudes were quantified as the average of 5 ms of the trace 65 ms after the stimulus artifact.

**Animals**. For the animal experiments presented in this study, mice heterozygous for a targeted loss-of-function mutation in the Ehmt1 gene ($Ehmt1^{+/−}$ mice) and their WT littermates on C57BL/6 background were used, as previously described[11]. Animal experiments were conducted in conformity with the Animal Care Committee of the Radboud University Nijmegen Medical Centre, The Netherlands, and conform to the guidelines of the Dutch Council for Animal Care and the European Communities Council Directive 2010/63/EU.

**Statistics**. The statistical analysis for all experiments was performed using GraphPad Prism 5 (GraphPad Software, Inc., CA, USA). We ensured normal distribution using a Kolmogorov–Smirnov normality test. To determine statistical significance for the different experimental conditions $p$ values < 0.05 were considered to be significant. Statistical analysis between lines were performed with two-ways ANOVA and Post hoc Bonferroni correction. We analyzed significance between Control and Kleefstra groups by means of the Mann–Whitney $U$ test. Data are presented as mean ± standard error of the mean (SE). Details about statistics are reported in Supplementary Data 1 (Excel file Statistics).

**Reporting summary**. Further information on research design is available in the Nature Research Reporting Summary linked to this article.

## Data availability

The datasets generated and/or analyzed during the current study are available from the corresponding author on reasonable request. Source data underlying Figs. 1–6 and Supplementary Figs. 1–8 is available as a Source Data file.

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

## Acknowledgements

We thank Dr. J. Ladewig for providing the NGN2 lentiviral construct. Funding: This work was supported by grants from: the Netherlands Organization for Scientific Research, open ALW ALW2PJ/13082 (to H.v.B., N.N.K., H.Z., T.K., and D.S.); grant 012.200.001 (to N.N.K.); the Netherlands Organization for Health Research and Development ZonMw grant 91718310 (to T.K.), 91217055 (to. H.v.B. and N.N.K.), SFARI grant 610264 (to N.N.K.) and the Jerome Lejeune Foundation (to H.v.B). GGA is supported by a postdoctoral research grant from TÜBITAK (1059B191600426). I.v.d.W is supported by the MINDED project, EU H2020 Marie Skłodowska-Curie (grant agreement No 754490).

## Author contributions

M.F., H.v.B., D.S., and N.N.K. conceived and supervised the study. M.N. performed all animal work. M.F., K.L., J.K., and N.N.K. designed all the experiments. M.F., K.L., J.K., G.G., B.M., J.v.R., K.F., N.K., A.O., W.v.d.A., I.v.d.W., T.K.G., and C.S. performed all the experiments. H.v.B., T.K., and H.Z. provided resources. D.S., M.F., K.L., G.G., N.K., K.F., A.O., B.M., J.v.R., and M.N. performed data analysis. M.F, K.L., J.K., and N.N.K. wrote the paper. D.S., H.v.B., T.K., and M.N. edited the paper.

## Competing interests

The authors declare no competing interests.
