## [Peer Review File · Nature Communications]

Reviewers' comments:

Reviewer #1 (Remarks to the Author):

This is an interesting paper from the Kasri group reporting that excess GRIN1/NR1 NMDA receptor expression in young glutamatergic neurons is a key driver for brain dysfunction in Kleefstra syndrome EHMT1 mutation-affected patient cell lines (NGN2-induced GLU neurons) and mouse mutant and in vivo model.

Given the implications as it pertains to opening novel therapeutic avenues, the paper is certainly extremely exciting for the field. However, there are still some significant issues that need additional experiments and clarifications:

a) while I am not an electrophysiologist, as far as I know, especially young neurons could assemble NMDA receptors that contain the GRIN3A/B NR3A/B subunit (replacing GRIN1) and such type of receptor could in some circumstances be more excitable and active than the receptor complex containing the GRIN1 subunit.

I could not find expression studies or any other experiment addressing this important issue. With other words, is the phenotype as reported by the authors caused by excess NR1-containing NMDA receptors or a shift in receptor composition with receptors carrying GRIN3 subunits???

b) related to a) because the Authors studied mixed cell culture systems (NGN2-induced glutamate neurons co-cultured with astrocytes, their immunoblots and RNA quantifications could all reflect excess GRIN1B expression in astrocytes. The authors should be easily able to rule this out.

Reviewer #2 (Remarks to the Author):

With this work the authors tried to elucidate the mechanisms by which epigenetic dysregulation of gene transcription affects neuronal network function. In particular, they focused on the Kleefstra syndrome, which is caused by mutations in histone methyltransferase EHMT1. The authors observed in IPS cells from patients with KS an alteration of the neuronal network and an upregulation of the NR1 subunit. In addition, they showed that NR1 up-regulation is due to a reduced deposition of the repressive catalytic product of EHMT1 at the GRIN1 promoter. Although the topic is interesting and the experimental results are quite comprehensive, the work presents several concerns which need to be addressed before publication.

Major points:

To be sure that the alterations are due to specific mutations the authors should always have the same genetic background so in all experiments should always be done using their isogenic controls.

- Fig1: The authors analyzed somatodendritic morphology finding no differences between controls and KS iNeurons. However, the dendritic spines density should be analyzed, considering the differences in dendritic spines highlighted in the Ehmt1+/- mice.

-

-Fig Suppl3: The authors state that at DIV21 both the KS controls and iNeurons show no differences in the active and passive properties of the membrane, however in Figure S3 (panel M) there is an increase in the AP decay time in the neurons of KS patients. How can this be explained? Can an alteration of potassium conductivity be excluded? Please discuss this point. Can mutations also change the properties of the resting membrane?

- Fig. S3: The sEPSCs demonstrating no changes except for the increased burst duration in KS iNeurons. However in figure S3 panel P, the amplitude of sEPSCs in KS iNeurons shows a strong

down-trend that, increasing the number of recorded neurons, could become significant. Considering the increased expression of NR1 subunit of NMDA receptors I would have expected an increased amplitude in KS iNeurons. The authors should clarify this point.

-

- Fig 4: In controls, APV treatment did not affect network burst activity but Nasp had a significant effect. How can this data be explained? If the hypothesis of an increased insertion of GluA2-lacking AMPA receptors after APV treatment in controls is right, I would have expected to see an increased burst activity. Could be the case that GluA2-containing AMPA receptors are exchanged with GluA2-lacking AMPA receptors?

-

- Fig 6: Could be interesting to see the AMPA/NMDA ratio in iNeurons. I suggest to perform this experiment.

- Could be interesting to try a behavioral rescue in the Ehmt1+/- mice using a NMDA antagonist treatment.

Minor points:

- The quality of figure 1E is not convincing, please do not only show the merge, but also the individual channels .

Panel S2 panel E , the KSiMOS image does not represent the quantification shown in the histogram (the staining of H3K9me2 is almost absent)

-also correct C2 with C1 in the histogram

also correct C2 with C1 in the histogram

- clarify in the text which synapsin was analyzed;

- Why in figure 4 panel C, D and figure 6 panel A you show normalized data instead of absolute values?

- Figure S3 panel P: "Amplitude (mV)" should be changed in "Amplitude (pA)"

Reviewer #3 (Remarks to the Author):

In the current manuscript, Frega et al. used human iPS cell derived induced neuronal cells to study the impact of mutations in EHMT1 gene on neuronal properties and gene expression. The experiments are well designed and the conclusion that EHMT1 haploinsufficiency leads to elevated NR1 expression, subsequent increased synaptic currents through NMDA receptor, and irregular network activity is interesting. While the conclusions from this work are largely consistent with the observations made in the rodent model of Kleefstra syndrome, the study seems rather incremental. However, as the first iPSC modeling of KS, this should not diminish the importance of the work.

1. In supplementary figure 3I, the arrows pointing to SYN1 and PSD95 puncta in the imaged image shifted.

2. Is the AMPAR/NMDAR ratio also changed in KS human iNs?

3. How did the authors define the bursts recorded from single neurons by patch-clamp? It is not clear in the supplementary figure 3P. Also, the recording was done using voltage clamp, but the data (amplitude) was presented using mV not currents.

4. In figure 2, when the authors analyzed the percentage of spike outside network burst, data from all different lines are used to generate the bar graph? It appears the percentage of outside spikes is not prominent in KS2, whereas there seem more outside spikes recorded from C2 compared to the two other control lines. It would be better to show the results using different control and KS lines as done for other parameters such as burst rate etc. And analyze the difference within each group (control vs. mutant cells).

5. In figure 3, the expression level of EHMT1 is about the RNA or protein?
6. In fig S5J, the size of nucleus of the mutant iNs seem smaller (almost half) that that of the control cells. This is a consistent phenotype or just a coincidence?
7. The authors showed loss of H3K9me2 occupancy at GRIN1 and BDNF promoter. Does EHMT1 haploinsufficiency cause genome wide transcription changes or only a subset of genes? If so, do these genes share some functional commonality? Or are regulated by a shared program? Are there evidences from mouse study to complement the current work?

Nael Nadif Kasri

Geert Grooteplein 21 Nijmegen
P.O. Box 9101, 6500 HB Nijmegen
The Netherlands

Telephone +31 243614242
n.nadif@donders.ru.nl

Nijmegen, August 2nd

Dear Reviewers

We are happy to send you our revised manuscript **NCOMMS-19-02277A-Z**, entitled: *Neuronal network dysfunction in a human model for Kleefstra syndrome mediated by enhanced NMDAR signaling* by Frega et al.

All changes in the manuscript are highlighted in red.

We made changes in the following figures:

- Figure 1E and 1F
- Figure 2O
- Figure 5A-E
- Figure 6A
- S2E
- S3G-O
- S5J
- S7E
- S8A-G

Point by point response to the comments of each of the reviewers is provided below:

Reviewer #1:

a) while I am not an electrophysiologist, as far as I know, especially young neurons could assemble NMDA receptors that contain the GRIN3A/B NR3A/B subunit (replacing GRIN1) and such type of receptor could in some circumstances be more excitable and active than the receptor complex containing the GRIN1 subunit. I could not find expression studies or any other experiment addressing this important tissue. With other words, is the phenotype as reported by the authors caused by excess NR1-containing NMDA receptors or a shift in receptor composition with receptors carrying GRIN3 subunits???

We thank the reviewer for his/her remark on *GRIN3A/B*. These NMDAR subunits have been less well studied but recently have been shown to play an important role during plasticity and behavior. In fact it has been shown that *GRIN3* subunits negatively regulate the contribution of NMDAR activity at synapses. As per suggestion of the reviewer, we have therefore included *GRIN3* in the qPCR analysis experiments (previous figure 5A). We could detect a signal for *GRIN3A* but not *GRIN3B* in our cell culture system. However, we observed no genotype differences for *GRIN3A* expression (**new Supplementary Figure S7E**).

b) related to a) because the Authors studies mixed cell culture systems (NGN2-induced glutamate neurons co-cultured with astrocytes, their immunoblots and RNA quantifications could all reflect excess GRIN1B expression in astrocytes. The authors should be easily able to rule this out.

In this study we used a mixed cell culture system, where Ngn2-induced glutamatergic neurons (iNeurons) are co-cultured with rat astrocytes. We were aware that the immunoblots and RNA quantifications could reflect the *GRIN1* expression in both neuron and astrocytes. In order to be able to quantify the *GRIN1* expression only in neurons, we designed human-specific primers. These were tested for human specificity, and showed no amplification when used on rat astrocytes only. Furthermore, we have now included new immunocytochemistry experiments showing increased NR1 in iNeurons. We are therefore convinced that the increased expression in *GRIN1* reflects an increase in the iNeurons as opposed to astrocytes. (**new fig 5C**)

Reviewer #2:

Major points:

- To be sure that the alterations are due to specific mutations the authors should always have the same genetic background so in all experiments should always be done using their isogenic controls.

We agree with the reviewer. To corroborate our results that the phenotypes observed in KS patient-derived neurons are due to a mutation in *EHMT1*, isogenic lines should be used. In this work we decided to use two pairs of isogenic lines: C_{MOS}/KS_{MOS} and C_{CRISPR}/KS_{CRISPR} . iPS cells were generated from a parent who had an affected child with KS due to a microdeletion on chromosome 9q34 (233 kb) including *EHMT1* that appeared present as a mosaic in the parent. From this same patient, we selected an iPS clone harboring the KS-causing mutation (KS_{MOS}) as well as a control clone not carrying the *EHMT1* deletion (C_{MOS}). This isogenic pair shares the same genetic background except for the KS-causing mutation, thereby reducing variability and enabling us to directly link phenotypes to heterozygous loss of *EHMT1*. To further address whether heterozygous loss of *EHMT1* is causal for the observed KS patient-derived network phenotypes, we generated a second set of isogenic iPS lines. We made use of CRISPR/Cas9 gene editing technology to generate an isogenic control and *EHMT1* mutant iPS cell line with a premature stop codon in exon 2 (respectively, C_{CRISPR} and KS_{CRISPR}).

As per suggestion of the reviewer we have now performed the functional experiments on all of our isogenic lines. These experiments included:

- quantification of synaptic puncta (Fig. 1E; 3C,F) and co-localization of Synapsin and PSD95 (**new Supplementary Figure S3G**)
- miniature excitatory postsynaptic currents (**new Supplementary Fig. S3H and Fig. 1F**)
- Spontaneous neuronal network activity on MEAs (Fig. 2 and Fig. 3)
- Effect of AMPAR and NMDAR blockade on network activity (Fig. 4 and S7)
- Effect of Regitabine and NASPM on network activity (Fig. 4 and S7)
- Increased NR1 expression (western blot, immunocytochemistry and H3K9me2 Chip experiments, **new Fig. 5A-C**)
- Increased NMDAR/AMPA ratio (**new Fig. 5D, E**)
- Rescue of network phenotypes by NMDAR antagonists (Fig. 7 and **new Fig S8**)

All these experiments were consistent with the observation that loss of *EHMT1* function leads to the neuronal phenotypes we describe here.

- Fig1: The authors analyzed somatodendritic morphology finding no differences between controls and KS iNeurons. However, the dendritic spines density should be analyzed, considering the differences in dendritic spines highlighted in the *Ehmt1*^{+/-} mice.

We thank the reviewer for his comment. Although dendritic spines can be visualized in iNeurons, the occurrence is extremely heterogeneous between cells and different batches. Given that in the mouse model (in the hippocampus) the changes in spine density were relatively modest and limited to only a fraction of the spines (stubby and mushroom) we chose not to analyze the dendritic spines. Rather we chose to analyze the functional correlates that are associated with structural changes, namely identification of functional synapses through immunocytochemistry (synapsin/PSD95 colocalization) and even more relevant, electrophysiological recordings of the excitatory synaptic strength, via the recording of miniature excitatory postsynaptic currents (mEPSC). This is currently the standard in the field and typically dendritic spines are not being analyzed in hiPSC-derived neurons, for the above-mentioned reasons.

Both, our immunocytochemical and electrophysiological measurements revealed no differences between genotypes. Furthermore we would like to emphasize that the changes in dendritic spines observed in the mouse models have only been observed in the hippocampus. Subsequent studies by us (Benevento et al. 2016; Martens et al., 2016) found no evidence for changes in excitatory synaptic strength (AMPA-mediated mEPSC), in the cortex or primary cortical neurons, which is closest to the cortical glutamatergic neurons (iNeurons) we generated in this study.

We have now included new data showing no difference in synapsin/PSD95 staining between genotypes (**new Supplementary Figure 3G**) and no difference in mEPSCs amplitude or frequency (**new Supplementary Figure**

3H).

-Fig Suppl3: The authors state that at DIV21 both the KS controls and iNeurons show no differences in the active and passive properties of the membrane, however in Figure S3 (panel M) there is an increase in the AP decay time in the neurons of KS patients. How can this be explained? Can an alteration of potassium conductivity be excluded? Please discuss this point. Can mutations also change the properties of the resting membrane?

Indeed, in general we observed no major differences in intrinsic properties between control and KS patient derived neurons. This can best be seen in the discriminative analysis plot, where we were unable to differentiate the different cell lines based on their intrinsic properties (**Suppl Fig. 6B**). However, when looking at individual parameters, we found an increase in the AP decay time compared to control iNeurons that was mainly driven by KS2 iNeurons. (**Table S3**). This increase in AP decay can indeed indicate a possible alteration in potassium conductivity/expression of voltage-gated potassium channels. Because we observed this mainly in one patient line, but not in the isogenic line, it could be an inherent property of this specific line.

The other possibility is that this reflects a mutation-specific effect. Of note, the KS2 line is the only line with a missense mutation in the pre-SET domain, with intact EHMT1 expression. It is thus plausible that within those KS2 iNeurons certain potassium channels are differently regulated, especially since we have previously shown that, in another model system of EHMT1 deficiency (shRNA knockdown) (Benevento et al., 2016), potassium channels were among the affected functional pathways. However, in this study, we did not pursue this angle further since the changes were not observed in the isogenic lines. We further thank the reviewer for noticing that we did not include all data on intrinsic properties, including the resting membrane data. We have now included a more comprehensive set of data **new Supplementary Fig. S3J-N**, describing the active and passive properties. We found no difference between genotypes for resting membrane and have now included these data (**new Supplementary Fig. S3N**). Of note, similarly to the AP decay we did also observe a line specific difference in Rheobase. We have therefore now rephrased our observation in the main text.

- Fig. S3: The sEPSCs demonstrating no changes except for the increased burst duration in KS iNeurons. However in figure S3 panel P, the amplitude of sEPSCs in KS iNeurons shows a strong down-trend that, increasing the number of recorded neurons, could become significant. Considering the increased expression of NR1 subunit of NMDA receptors I would have expected an increased amplitude in KS iNeurons. The authors should clarify this point.

We thank the reviewer for this observation. To determine whether there would be any significant difference we now increased the number of recorded neurons, also adding one isogenic line (C_{MOS} and KS_{MOS}) to the dataset. We did not obtain significance, rather we found the contrary: the additional data points increased the P value from 0.06 to 0.077 (**new Supplementary figure S3O and Table S3**).

One should note that these are spontaneous events recorded at -60 mV in the absence of TTX. Amplitude and frequency of such events therefore cannot be directly correlated to the amount of AMPA (or NMDA) receptors since we are looking at network activity. To determine directly the amount of functional NMDARs at the synapse we have now included new single-cell recording experiments where we measured the NMDAR/AMPA ratio after stimulation (see comments below, **new fig 5D,E**). These data do show an increase in NMDAR currents.

- Fig 4: In controls, APV treatment did not affect network burst activity but Nasp^m had a significant effect. How can this data be explained? If the hypothesis of an increased insertion of GluA2-lacking AMPA receptors after APV treatment in controls is right, I would have expected to see an increased burst activity. Could be the case that GluA2-containing AMPA receptors are exchanged with GluA2-lacking AMPA receptors?

Indeed, in the original manuscript we already reported on this observation: *“Of note, although in control networks burst activity was not suppressed by D-AP5, we did observe that Nasp^m had a small but significant effect on the burst rate. This indicates that NMDAR blockage in controls also induced synaptic insertion of GluA2-lacking AMPARs, albeit less pronounced than in KS networks.”*

We observed that burst activity in controls was not as suppressed or significantly affected by D-APV treatment

compared to KS cells. Nevertheless, we did observe a slight decrease in the burst activity (about 10-20% decrease compared to baseline) in controls, as shown in **Figure 4E and S7A**. It is likely that network activity in controls usually includes a partial activation of NMDARs, but their contribution to network bursts is much less compared to KS networks.

Our hypothesis is that NMDA receptor blockade induces plasticity not only in KS but also in control networks, as reported by Lee et al., (2012). We think that the amount of GluA2-lacking AMPARs inserted at the synapse depends proportionally on the magnitude of the burst suppression, and this is a homeostatic response to global changes in network activity. A more dramatic effect was observed when activity was completely silenced by Retigabine. This is also what we see when bursts are suppressed by Retigabine, independent of direct NMDAR blockade. In this case both Control and Ks respond equally to Retigabine and Nasp. Our data would favor a model in which GluA2-containing receptors are exchanged by GluA2-lacking receptors in controls. We have now discussed this in the Discussion section.

- Fig 6: Could be interesting to see the AMPA/NMDA ratio in iNeurons. I suggest to perform this experiment. This is indeed a very good suggestion. We have now included these data. Specifically, we implemented a new set of experiments in which iNeurons were infected (approx. 80%) with AAV-mCherry-ChR2 (Channelrhodopsin). Non-infected cells were patched and ChR2-infected cells were stimulated with blue light to elicit glutamate release. We measured AMPAR-mediated responses at -70mV and NMDAR-mediated responses at +40mV for each cell and calculated the NMDAR/AMPA ratio. For all isogenic lines we find an increase in NMDAR/AMPA ratio, with no change in AMPAR mediated currents (additional mEPSC recordings, **see fig. S3H**). These data further corroborate our observations that NMDAR function is increased in Kleefstra syndrome patient-derived neurons compared to controls (**new fig. 5D,E**).

- Could be interesting to try a behavioral rescue in the Ehmt1+/- mice using a NMDA antagonist treatment. We agree that attempting to rescue behavior by treating the mouse model with an NMDAR antagonist would be interesting. However, to be done properly and carefully, this would require an extensive amount of work, which we find to be out of the scope of this study. In particular, an *in vivo* drug study would require us to, first, determine relevant behavioral endpoints to evaluate, determine a proper treatment regimen to follow (e.g., dose-response, acute vs. chronic treatment) and also the developmental time window where altered NMDAR activity is most relevant. In fact, several recent studies have been dedicated solely to testing NMDAR antagonists for their potential efficacy in rescuing behavioral deficits in mouse models of NDDs (Chung et al., 2019 Biol Psych; Tang et al., 2019 Nat Comm; Tu et al., 2017 Nat Comm; Patrizi et al., 2016 Biol Psych). It is clear from these studies that NMDAR hyperactivity can occur in specific brain regions, at different time points during development, and can affect different types of behavior and thus require extensive experimental testing. We have, however, now elaborated in the Discussion on relevant follow-up experiments, especially in light of the aforementioned literature.

Minor points:

- The quality of figure 1E is not convincing, please do not only show the merge, but also the individual channels. We changed Figure 1E based on the suggestion of the reviewer.

- Panel S2 panel E, the KSMOS image does not represent the quantification shown in the histogram (the staining of H3K9me2 is almost absent)
We changed the image in Figure S2E to a more representative one.

-also correct C2 with C1 in the histogram

Indeed, we made a mistake in the figure. The experiments were conducted on C₂, C_{MOS}, KS₁, KS₂ and KS_{MOS}, so the names of the cell lines shown in the histogram are correct. However, the images shown in the first column of panel E are relative of C₂. Thus, we changed the names accordingly.

- clarify in the text which synapsin was analyzed;

We have now indicated in the text that we used an antibody against Synapsin1/2 (guinea pig anti-synapsin1/2; 1:1000; Synaptic Systems 106004)

- Why in figure 4 panel C, D and figure 6 panel A you show normalized data instead of absolute values?

In figure 4 panel C,D we decided to use normalized values to show the effect of D-APV on the burst frequency. The absolute values for C_{MOS} and KS_{MOS} are already shown in figure 2J. Because there is a basal difference between the level of bursting activity exhibited by control and KS neuronal networks, we decided to normalize the burst frequency after NMDAR blockade against the burst frequency before treatment. We think that in this way it is possible to better observe the variation in frequency and to compare the effect of the drug (also over time, figure 4E and on different cell lines, figure S7A-D) on a parameter that shows a significant difference between C_{MOS} and KS_{MOS} prior to treatment.

In figure 6 we show normalized values for firing rate, burst rate and burst duration taken from our mouse model. We agree with the reviewer that absolute values give more information compared to normalized values. We therefore adjusted the figure accordingly (**adapted fig. 6A**). Previously, values from $Ehmt1^{+/-}$ mice were normalized against $Ehmt1^{+/+}$ mice from the same litter. In the new analysis, absolute values from all experiments were averaged. For this reason, we can observe small differences with the previous normalized graphs. However, the statistical differences in the burst frequency and duration persist.

- Figure S3 panel P: "Amplitude (mV)" should be changed in "Amplitude (pA)"

We thank the reviewer for noticing this mistake. Indeed, the amplitude should be in pA not in mV. We have changed it in the figure (**new fig. S3N**).

Reviewer #3:

1. In supplementary figure 3I, the arrows pointing to SYN1 and PSD95 puncta in the imaged image shifted.

We have adjusted the figure accordingly. We have also included a quantification of co-localization (**new Fig. S3G**)

2. Is the AMPAR/NMDAR ratio also changed in KS human iNs?

This is indeed a very good suggestion. We have now included these data. Specifically, we implemented a new set of experiments in which iNeurons were infected (approx. 80%) with AAV-mCherry-ChR2 (Channelrhodopsin). Non-infected cells were patched and ChR2-infected cells were stimulated with blue light to elicit glutamate release. We measured AMPAR-mediated responses at -70mV and NMDAR-mediated responses at +40mV for each cell and calculated the NMDAR/AMPA ratio. For all isogenic lines we find an increase in NMDAR/AMPA ratio, with no change in AMPAR mediated currents (additional mEPSC recordings, **see fig. S3H**). These data further corroborate our observations that NMDAR function is increased in Kleefstra syndrome patient-derived neurons compared to controls (**new fig. 5D,E**).

3. How did the authors define the bursts recorded from single neurons by patch-clamp? It is not clear in the supplementary figure 3P. Also, the recording was done using voltage clamp, but the data (amplitude) was presented using mV not currents.

We have now added in the Material and Methods section our definition of burst detection from single neurons: *"Bursts were detected when at least 5 events with a minimal inter-event-interval of 100 ms were exhibited by a cell. Burst detection, burst frequency and duration were computed with software developed in MATLAB (The Mathworks, Natick, MA, USA)."*

In Supplementary figure 3P (**new Supplementary Fig. S3N**) the amplitude should be in pA not in mV. We have corrected this in the figure.

4. In figure 2, when the authors analyzed the percentage of spike outside network burst, data from all different lines are used to generate the bar graph? It appears the percentage of outside spikes is not prominent in KS2, whereas there seem more outside spikes recorded from C2 compared to the two other control lines. It would be better to show as the results using different control and KS lines as done for other parameters such as burst rate etc. And analyze the difference within each group (control vs. mutant cells).

Indeed, in figure 2O data from all the different cell lines were used to generate the bar graph. As suggested by the reviewer, we now show the results for each individual cell line, indicating the values for each experiment, as well as for the other parameters (**new fig. 2O**). With this visualization, it is possible to observe that KS cell

lines show a statistically lower percentage of spikes outside the bursts compared to controls. The statistical differences are reported in **Table S3**.

5. In figure 3, the expression level of EHMT1 is about the RNA or protein?

In figure 3E the graph shows the quantification of relative EHMT1 protein level. For clarity, we changed the figure legend as follow:

“E) Quantification of relative EHMT1 protein level,”.

6. In fig S5J, the size of nucleus of the mutant iNs seem smaller (almost half) that that of the control cells. This is a consistent phenotype or just a coincidence?

We thank the reviewer for this comment. We calculated the size of nuclei for C_{CRISPR} and K_{S_{CRISPR}} but found no statistical difference between them. The values are shown in **new Supplementary Figure S5J**.

7. The authors showed loss of H3K9me2 occupancy at GRIN1 and BDNF promoter. Does EHMT1 haploinsufficiency cause genome wide transcription changes or only a subset of genes? If so, do these genes share some functional commonality? Or are regulated by a shared program? Are there evidences from mouse study to complement the current work?

We and others have indeed previously shown in cellular and mouse models that EHMT1 haploinsufficiency can cause transcriptional changes beyond NR1 and BDNF. In general these changes were found to be relatively modest compared, especially to what could be expected from a chromatin remodeler (limited to a few hundreds genes) (Schaefer et al, Benevento et al., Iacono et al., Frega et al.,). Direct extensive comparison with and between these data is probably difficult due the heterogeneity of the studied material (many different cell types in mouse material) and the use of different model system (KO, conditional KO, shRNA) and brain regions. Nevertheless it is interesting to notice pathways related to synapse function and ion channel activity were found to be affected. This included the differential H3K9me2 occupancy and expression of BDNF, which we showed to be attributed to altered in vitro and in vivo homeostatic plasticity in Ehmt1-lacking primary neurons and mice (Benevento et al.), but also, among others, increased NR1 expression in Ehmt1 knockdown rat primary cultures (Frega et al.). We thank the reviewer for pointing this out, we have now briefly referred to these studies in the discussion.

Sincerely Yours

Dr. Nael Nadif Kasri

References

Benevento, M., et al. (2016). Histone Methylation by the Kleefstra Syndrome Protein EHMT1 Mediates Homeostatic Synaptic Scaling. *Neuron*, 1–16. <http://doi.org/10.1016/j.neuron.2016.06.003>

Chung et al., Early Correction of N-Methyl-D-Aspartate Receptor Function Improves Autistic-like Social Behaviors in Adult Shank2^{-/-} Mice. (2019), *85*(7), 534–543. <http://doi.org/10.1016/j.biopsych.2018.09.025>

Frega, M., et al. (2018). Distinct pathogenic genes causing intellectual disability and autism exhibit overlapping effects on neuronal network development, 1–41. <http://doi.org/10.1101/408252>

Lee, H.-K. (2012). Ca-permeable AMPA receptors in homeostatic synaptic plasticity. *Frontiers in Molecular Neuroscience*, 5, 17. <http://doi.org/10.3389/fnmol.2012.00017>

Martens, M. B., et al. (2016). Euchromatin histone methyltransferase 1 regulates cortical neuronal network development. *Scientific Reports*, 1–11. <http://doi.org/10.1038/srep35756>

Patrizi et al., Chronic Administration of the N-Methyl-D-Aspartate Receptor Antagonist Ketamine Improves Rett Syndrome Phenotype. (2016). 79(9), 755–764. <http://doi.org/10.1016/j.biopsych.2015.08.018>

Schaefer, A., Sampath, S. C., Intrator, A., Min, A., Gertler, T. S., Surmeier, D. J., et al. (2009). Control of cognition and adaptive behavior by the GLP/G9a epigenetic suppressor complex. *Neuron*, 64(5), 678–691. <http://doi.org/10.1016/j.neuron.2009.11.019>

Tang et al., Altered NMDAR signaling underlies autistic-like features in mouse models of CDKL5 deficiency disorder. (2019). 10(1), 2655. <http://doi.org/10.1038/s41467-019-10689-w>

Tu et al., NitroSynapsin therapy for a mouse MEF2C haploinsufficiency model of human autism. (2017). 8(1), 1488. <http://doi.org/10.1038/s41467-017-01563-8>

REVIEWERS' COMMENTS:

Reviewer #1 (Remarks to the Author):

The Authors appear to have carefully and properly responded to previous review. I have no additional concerns.

Reviewer #2 (Remarks to the Author):

The authors have fully addressed my comments. I have no more question.

Reviewer #3 (Remarks to the Author):

In the revised manuscript, the authors have adequately addressed the reviewer's previous comments. One thing that is worth pointing out in the manuscript is in the 233kb microdeletion, is there any other coding sequence deleted in addition to EHMT1? For some of changes including burst rate and duration, there seems a trend of difference between the KSmos and other patient lines. Is it because of other sequences affected in the deleted region?

Nael Nadif Kasri

Geert Groteplein 21 Nijmegen
P.O. Box 9101, 6500 HB Nijmegen
The Netherlands

Telephone +31 243614242
n.nadif@donders.ru.nl

Nijmegen, September 14th

We are very happy read to you have accepted our manuscript **NCOMMS-19-02277B**, entitled: *Neuronal network dysfunction in a human model for Kleefstra syndrome mediated by enhanced NMDAR signaling* by Frega et al. for publication in Nature Communications.

Below the response to the remaining point of reviewer#3:

Reviewer #3:

In the revised manuscript, the authors have adequately addressed the reviewer's previous comments. One thing that is worth pointing out in the manuscript is in the 233kb microdeletion, is there any other coding sequence deleted in addition to EHMT1? For some of changes including burst rate and duration, there seems a trend of difference between the KSmos and other patient lines. Is it because of other sequences affected in the deleted region?

We thank the reviewer for his/her remark. We have now better described the microdeletion (which also included a part of the *CACNA1B* gene) and included this information in the discussion and methods part of the paper.

Sincerely Yours

Dr. Nael Nadif Kasri